# Exposing and Addressing Cross-Task Inconsistency in Unified Vision-Language Models

**Adyasha Maharana**                                                          *adyasha@cs.unc.edu*
*University of North Carolina, Chapel Hill*

**Amita Kamath**                                                                *kamatha@cs.ucla.edu*
*University of California, Los Angeles*

**Christopher Clark**                                                           *chrisc@allenai.org*
*Allen Institute for AI*

**Mohit Bansal**                                                                *mbansal@cs.unc.edu*
*University of North Carolina, Chapel Hill*

**Aniruddha Kembhavi**                                                          *anik@allenai.org*
*Allen Institute for AI*

**Reviewed on OpenReview:** *https://openreview.net/forum?id=ue9igTDLN2*

## Abstract

As general purpose vision models get increasingly effective at a wide set of tasks, it is imperative that they be consistent across the tasks they support. Inconsistent AI models are considered brittle and untrustworthy by human users and are more challenging to incorporate into larger systems that take dependencies on their outputs. Measuring consistency between very heterogeneous tasks that might include outputs in different modalities is challenging since it is difficult to determine if the predictions are consistent with one another. As a solution, we introduce a benchmark dataset, CocoCon, where we create contrast sets by modifying test instances for multiple tasks in small but semantically meaningful ways to change the gold label and outline metrics for measuring if a model is consistent by ranking the original and perturbed instances across tasks. We find that state-of-the-art vision-language models suffer from a surprisingly high degree of inconsistent behavior across tasks, especially for more heterogeneous tasks. To alleviate this issue, we propose a rank correlation-based auxiliary training objective, computed over large automatically created cross-task contrast sets, that improves the multi-task consistency of large unified models while retaining their original accuracy on downstream tasks. Data and code are available at `https://adymaharana.github.io/cococon/`.

## 1 Introduction

General Purpose Vision (GPV) models (Gupta et al., 2022a; Kamath et al., 2022; Cho et al., 2021; Lu et al., 2022; Wang et al., 2022) are trained to perform many diverse multimodal tasks ranging from visual question answering (VQA) and referring expression grounding to semantic segmentation and image generation. A fundamental requirement and intuitive expectation of such systems is that they provide consistent results across the tasks they support. For example, if a system produces the caption *two jaguars are sitting on a tree branch* then one would expect it to answer the question *What animals are these?* with *jaguars* and to return two bounding boxes if asked to locate the *jaguars*.

While the latest GPV models (Lu et al., 2022; Wang et al., 2022; Huang et al., 2023) perform impressively on multi-task benchmarks (Gupta et al., 2022b), we find that these models can provide surprisingly inconsistent

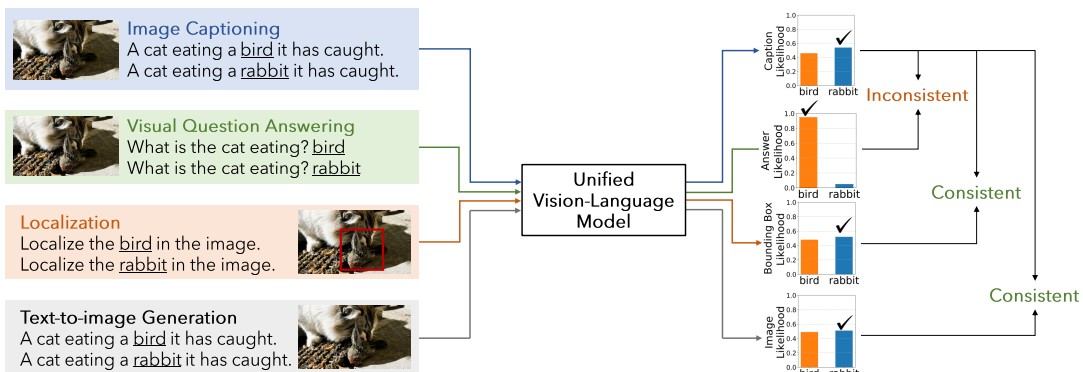

Figure 1: Examples of consistent and inconsistent predictions from Unified-IO$_{XL}$ (Lu et al., 2022).

answers for simple images and tasks. Fig. 1 shows an example where Unified-IO$_{XL}$ (Lu et al., 2022) prefers the caption: *A cat eating a rabbit it has caught*, but then answers *bird* when asked *What is the cat eating?* Solving multiple tasks for one image may require some degree of specialized reasoning, but they necessitate a semantic interpretation of the input image which should be common across tasks. When models demonstrate such trivial inconsistencies, it is hard for end users to trust them, particularly in important applications, because it is harder to understand and predict their behavior. From a practical standpoint, it is challenging to incorporate such models into larger systems, because it's hard to calibrate for them. Finally, from a philosophical view, having different interpretations of an image depending on the target task defies how we intuitively think unified models should behave.

In computer vision, cross-task consistency has been of some interest for classical tasks (Zamir et al., 2020), while in natural language processing past work has studied consistency for tasks like question-answering (Kassner et al., 2021). However, in vision-and-language research, much work has focused on within-task consistency for visual question answering (Shah et al., 2019; Ribeiro et al., 2019; Dharur et al., 2021; Ray et al., 2019; Bitton et al., 2021). Semantic consistency of multi-modal models *across tasks* has remained unexplored, partly due to the absence of models (until recently) that can perform various tasks simultaneously and effectively.

With recent advances in GPV research, we can now probe models for cross-task consistency. A simple and straightforward method is to compute the semantic overlap between a model's predictions for the same image across tasks. While possible for related tasks like captioning and VQA, measuring semantic overlap between outputs from different modalities can be ill-defined (e.g. it is unclear how to quantify the overlap between bounding boxes for localization and an answer for VQA). Additionally, models may perform well by producing simple outputs for tasks. For example, if a model generates short captions about a single subject, this method can only probe consistency for that narrow set of visual elements. Instead, we choose to utilize human-defined outputs for tasks that cover a wide range of semantic elements for a complete evaluation of consistency. For a given pair of tasks, we perturb the test instances in similar but meaningful ways that change the gold label, in order to create contrast sets (Gardner et al., 2020). More likely perturbations (e.g. *keyboard → laptop* in Fig. 2(b)) lead to harder contrast sets whereas less likely perturbations (e.g. *keyboard → earbuds*) lead to easier contrast sets. Then, we measure a model's likelihood of predicting the ground truths as well as their contrast counterparts for both tasks. If a model is more likely to predict the contrast output for one task and the ground truth output for the other task or vice-versa, it implies that the model has contradicting interpretations of the same input for the two tasks. In the example shown in Fig. 2, the model favors the caption with *computer keyboard*, but is more likely to answer *laptop* in response to the question: *What is the object in the lower right-hand corner?*, leading to cross-task inconsistency. Operating with likelihoods also allows us to overcome the challenges of comparing outputs from two different modalities.

For this purpose, we present CocoCon, a benchmark dataset with contrast sets for four commonly used multimodal tasks. Each sample in CocoCon contains up to five contrast sets of varying difficulty for each of the tasks. We use image captioning as an *anchor task* because captions contain semantic elements used by most other tasks and evaluate it against VQA which has textual outputs, localization which has bounding

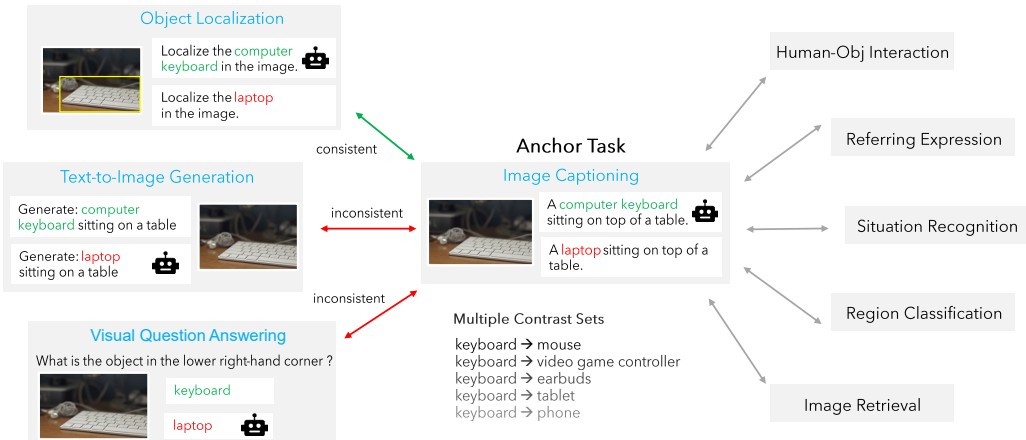

Our evaluation framework for probing cross-task consistency in unified models via contrast sets.

Figure 2: **Illustration of our method for probing inconsistencies across tasks.** We build candidate answers for multiple tasks that correspond to different semantic understandings of an image (e.g., keyboard vs. laptop), and check if the model's preferred answers across tasks match the same semantic understanding.

box outputs, and image generation with image outputs. This covers task pairs with outputs in the same output modalities as well as different output modalities. We measure consistency % as well as Spearman's rank correlation coefficient between the ranking of contrast sets.

We evaluate two recent GPV models, Unified-IO (Lu et al., 2022) and OFA (Wang et al., 2022), both of which support all four tasks in CocoCon. Additionally, we evaluate Kosmos-2 (Peng et al., 2024) and GILL (Koh et al., 2023) which support three out of four tasks in CocoCon. We show that cross-task inconsistency is a surprisingly significant phenomenon in these models across all tasks in CocoCon and various model sizes. Inconsistency increases with the heterogeneity between output modalities within a pair of tasks as well as with the complexity of the tasks themselves. Moreover, consistency improves with easier contrast sets, yet remains significantly less than 100% for all tasks. We also find that larger models are more consistent by virtue of being more accurate at the tasks. Finally, our evaluation suggests that multi-task models capable of performing a larger set of tasks are more inconsistent.

Cross-task inconsistency is undesirable in a unified model, and it is paramount that we work toward mitigating it. To this end, we propose using a consistency objective utilizing large automatically generated cross-task contrast sets and a rank correlation loss objective via soft ranking (Blondel et al., 2020). Our experiments show that continued training of models using this auxiliary consistency-based objective can lead to consistency improvements when evaluated on CocoCon while preserving or improving the accuracy of the model on the original test sets. In summary, our contributions include:

(a) highlighting the issue of cross-task inconsistency in multi-modal models,

(b) introducing the use of contrast sets and a benchmark dataset, CocoCon, to measure cross-task inconsistency amongst four popular multimodal tasks,

(c) demonstrating the inconsistent behavior of state-of-the-art vision-language models, and

(d) a consistency-based training objective to improve consistency without compromising accuracy.

## 2 Related Work

To our knowledge, no existing work evaluates cross-task consistency for multi-modal models. In this section, we discuss studies that evaluate and enforce consistency for individual or multiple tasks within one modality.

**Consistency for VQA.** Shah et al. (2019) revealed that VQA models are inconsistent across linguistic variations of a visual question, then improved consistency using automatic data augmentation; an approach which was further improved in Kant et al. (2021) using an additional contrastive loss. Ribeiro et al. (2019);

Ray et al. (2019) evaluated consistency across the original QA data and automatically generated QA pairs implied by this data. Selvaraju et al. (2020) collected human-annotated sub-questions to evaluate model reasoning capabilities through the lens of consistency. Dharur et al. (2021) trained models to rank the sub-questions proposed by SQuINT (Selvaraju et al., 2020) higher than unrelated questions from the same image, making models more consistent across both sub-questions and rephrasings of the question. Contrast sets have also been used to measure and improve consistency for VQA (Ribeiro et al., 2019; Bitton et al., 2021). Unlike these works, our approach evaluates and improves consistency across multiple tasks.

**Consistency for NLP.** Consistency has also been discussed in NLP, primarily in the single-task setting. Elazar et al. (2021) evaluated and improved factual consistency of pre-trained LMs across paraphrasings of factual statements. Kassner et al. (2021) considered the responses of a pre-trained LM to a stream of questions, and evaluated and improved the consistency and accuracy of its answers over time. Kaushik et al. (2019) collected counterfactual instances to evaluate the overreliance of NLP models on spurious attributes. Gardner et al. (2020) manually created contrast sets for 10 individual NLP tasks to evaluate single-task consistent responses across meaning-altering perturbations. In comparison to these works, we evaluate consistency across multiple tasks, without the need for human annotations as used in Gardner et al. (2020). Nishino et al. (2019) used multi-task learning with a hierarchical consistency objective to predict the headlines, key phrases, and categories of articles; however, the model uses separate decoders per task. Our work studies cross-task consistency of General Purpose Vision (GPV) models with unified output decoders.

**Cross-task Consistency for Vision.** Cross-task relationships among classic vision tasks have been studied by Zamir et al. (2018). Lu et al. (2021) used geometry and physics to identify consistency constraints between such tasks, and use them to improve performance in low data regimes. Zamir et al. (2020) enforced cross-task consistency for vision tasks using inference-path invariance and demonstrate their method for tasks in the pixel space (like depth and surface normals). It is not straightforward to extend this approach to vision and language tasks which are often conditioned not just on an image but also on a language input and where one task's output may not easily be transformed into another's output.

## 3 Contrast Sets for Cross-Task Consistency

In this section, we describe the problem of inconsistency across tasks in unified models, motivate the use of contrast sets to evaluate consistency, and outline our framework for measuring cross-task consistency.

**The Problem.** In the pursuit of developing task- and modality-agnostic unified systems, models like Unified-IO (Lu et al., 2022) are trained on a variety of tasks geared towards learning robust semantic representations of the input. Each task is designed to strengthen the model's understanding of a distinct perspective of the ground truth. For instance, a visuo-linguistic model is simultaneously trained to generate a caption for the entire image as well as answer questions about subjects in the image. The popular and effective training paradigm for such models is to learn a probability distribution over the space of possible outputs and maximize the likelihood of the target output. This leads to an inherent ranking of possible outputs based on their probabilities, which can be used to rank outputs that reflect distinct semantic understandings of the input. For a reliable and truly unified model, the ranked space of such probable outputs should also be aligned *across tasks*. However, (see Fig. 1), we find that unified models can interpret inputs differently for different tasks, leading to misalignment between these spaces and inconsistency in predictions. We measure this inconsistency with the help of contrast sets.

**Contrast Sets.** Model performances on the i.i.d. test data are often treated as an absolute measurement of its abilities. However, when the test data has systematic gaps like annotation artifacts (Gururangan et al., 2018), the model can learn simple decision boundaries to solve the dataset and result in misleading high performances. Gardner et al. (2020) introduce contrast sets to close such systematic gaps in evaluation. Contrast sets are created by perturbing test instances in meaningful ways that change the gold label. This allows for the evaluation of a model's local decision boundary around a *pivot* test instance and measurement of how well it *aligns with the correct decision boundary*. Models with simple decision boundaries fail to perform well on contrast sets. Using the same intuition, we can create equivalent perturbations on a test instance for a pair of tasks and evaluate whether the unified model performs similarly on the contrast set

for either task. In this manner, we leverage the framework of contrast sets to measure how well a model's decision boundaries for two distinct tasks *align with each other*.

Consider a model with parameters $\theta$ and two tasks $t_0$, $t_1$ that can be performed by the model. To construct a contrast set, we first pick a test instance and the respective ground truth annotations for each task i.e. $(x_{t_0}, y_{t_0})$, $(x_{t_1}, y_{t_1})$, termed as the **pivot** instances. We define the space of contrast outputs for an instance $x$ as the set of outputs $\tilde{y}$ that are within some distance $\epsilon$ of $y$. That is, $C(x) = \{(\tilde{y} \,|\, d(y, \tilde{y}) < \epsilon\}$, where $d(.)$ is some distance function. Let $f_\theta(y|x)$ be the likelihood of model $\theta$ for predicting the output $y$ in response to input $x$. Now, we define the model $\theta$ to be consistent across tasks $t_0, t_1$ with respect to the pivots $x_{t_0}, x_{t_1}$ if the model is more likely to predict the gold outputs $y_{t_0}, y_{t_1}$ in both tasks, as compared to their respective contrast outputs $\tilde{y}_{t_0}, \tilde{y}_{t_1}$. The model is also considered consistent if it assigns a larger likelihood to the contrast outputs than the gold outputs of both tasks because, even if the model answers wrongly for both tasks, as long as it reflects a common understanding of the input, the model is consistent. Mathematically,

$$\mathcal{C} = \begin{cases} 1 & \text{if } f_\theta(y_{t_0}|x_{t_0}) > f_\theta(\tilde{y}_{t_0}|x_{t_0}) \ \wedge \ f_\theta(y_{t_1}|x_{t_1}) > f_\theta(\tilde{y}_{t_1}|x_{t_1}) \\ 1 & \text{if } f_\theta(y_{t_0}|x_{t_0}) < f_\theta(\tilde{y}_{t_0}|x_{t_0}) \ \wedge \ f_\theta(y_{t_1}|x_{t_1}) < f_\theta(\tilde{y}_{t_1}|x_{t_1}) \\ 0 & \text{otherwise} \end{cases} \quad (1)$$

where $\tilde{y}_{t_0} \in C(x_{t_0})$, $\tilde{y}_{t_1} \in C(x_{t_1})$ and $\mathcal{C}$ is the consistency score. This framework can be easily extended to more than two tasks. For the scenario of $> 2$ tasks, we define an **anchor task** $t_0$, that contains semantic elements common to each of the remaining tasks. Then, we compute pairwise consistency scores for the anchor and the rest of the tasks $\{t_1, t_2, \ldots, t_T\}$ i.e. we have $T$ pairwise scores for $T$ tasks.

**Difficulty ($k$).** Contrast sets can be of varying difficulty, depending on the likelihood of the perturbations used to create the contrast sets. For example, *basketball* is a likelier substitute for the semantic concept *football* whereas *kite* is much less likely. Hence, the contrast set containing *basketball* is a **hard** contrast set and the one containing *kite* is an **easy** contrast set. We rank all contrast sets for a given instance and use the rank $k$ to indicate the difficulty i.e. lower $k$ implies harder contrast sets.

**Evaluation Metrics.** We introduce two metrics for calculating the consistency of a model over a dataset of $N$ samples, containing $K$ contrast sets each, for $T$ tasks. Each sample consists of pivot instances for the $T$ tasks and the corresponding sets of up to $K$ contrast outputs. We first rank the $K$ contrast sets by difficulty according to the model's likelihoods for the anchor task, $\{\tilde{y}_{t_0}^1, \ldots \tilde{y}_{t_0}^K\}$. For each task $t_i$ and at each $k$, we compute **% consistency @ $k$ ($\mathcal{C}_k$)** as the % of samples for which the model is consistent i.e.,

$$\mathcal{C}_k = \frac{1}{N}\sum_{i=1}^{N} \mathcal{C}(y_{t_0}, y_{t_i}, \tilde{y}_{t_0}^k, \tilde{y}_{t_i}^k) \quad (2)$$

where consistency $\mathcal{C}(.)$ is computed as per Eqn. 1. Higher values for $\mathcal{C}_k$ suggest that the model is more consistent across $t_0$ and $t_i$. This metric measures consistency with respect to the ground truth annotations, which are used as pivots in our setup. We also compute **spearmanr** ($\rho_{rank}$), the Spearman's rank correlation coefficient over the ranked contrast outputs for both tasks, in order to measure the global alignment between the two output spaces. We observe these metrics in tandem with task-specific accuracies to avoid overestimating a model with degenerate but consistent solutions.

## 4 The CocoCon Benchmark

In this section, we detail the construction and composition of our benchmark dataset CocoCon, developed as per the framework in Sec. 3. Then, we discuss the statistics and evaluation framework of CocoCon.

### 4.1 Dataset Construction

The COCO dataset (Lin et al., 2014) contains annotations for many tasks in vision and language, which makes it very suitable for evaluating cross-task consistency in a multimodal model. CocoCon is created from the validation splits of the four tasks i.e. image captioning (**anchor task**), VQA (Antol et al., 2015; Goyal et al., 2017), localization, and text-to-image generation. The dataset creation pipeline is as follows.

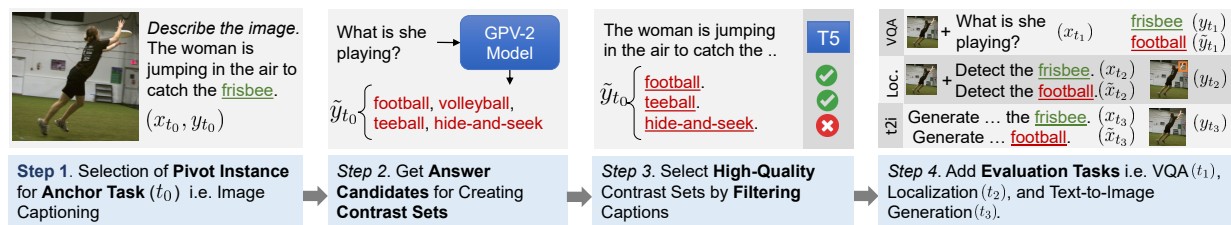

Figure 3: **Step-by-step demonstration of the automated pipeline for generating contrast sets.** Contrast sets generated from this pipeline are manually filtered to prepare the CocoCon benchmark.

**(Step 1) Selection of Pivot Instances.** First, we select **pivot instances** for the captioning and VQA tasks since it is easy to compute semantic overlap between the outputs of these tasks. Existing captioning annotations for the COCO dataset were filtered to retain ones that had semantic overlap with at least one question-answer pair from VQAv2 annotations. For instance, the caption: *The woman is jumping in the air to catch the frisbee.* from COCO overlaps with the VQA sample: *What is she playing? frisbee* (see Fig. 3, Step 1) and was retained in our method. The semantic overlap was computed using a series of text-processing steps including lemmatization and word overlap.

**(Step 2) Contrast Set Candidates.** Next, we need to substitute the overlapping semantic concept with other likely concepts to create contrast sets. There are many ways to perform this step. For instance, these perturbations can be written by human annotators, which might result in undesirable systematic biases in the contrast sets (Gururangan et al., 2018). Adversarial methods advocate gradient-based methods to get hard negatives for such perturbations (Alzantot et al., 2018), however, we want to avoid integrating the biases of the models we are evaluating into a benchmark dataset.

In contrast, we choose to use probable answers to the VQA questions from an off-the-shelf VQA model, GPV-2 (Kamath et al., 2022), to create a large set of perturbations (see Fig. 3, Step 2). GPV-2 is trained on the Web10K dataset (Kamath et al., 2022) that contains semantic concepts beyond COCO. This makes the contrast sets in CocoCon diverse and additionally challenging for unified models. Note that we do not evaluate GPV-2 on CocoCon since it can perform only a subset of the tasks present in it (see Fig. 2).

**(Step 3) Filtering.** The perturbations obtained from the previous step are filtered to retain high-quality candidates only, by creating contrast captions and retaining captions (and the corresponding contrast VQA samples) with high scores from the T5 language model i.e., the ungrammatical and nonsensical captions are filtered out. For instance, in Fig. 3 (see Step 3), the GPV-2 answer *hide-and-seek* is filtered out using T5 score, because *catch the hide-and-seek* is an unlikely phrase.

**(Step 4) Heterogeneous Evaluation Tasks.** The next step is to add evaluation tasks with heterogeneous output modalities i.e., localization and text-to-image generation (see Fig. 3, Step 4). For the localization task, the automatically generated dataset from the last step is merged with the COCO localization annotations. Annotations for localization in COCO images pertain to the narrow set of pre-defined COCO objects, which may or may not appear in the caption. Only those objects which appear in the caption and VQA answer are retained in CocoCon for the localization task. The contrast outputs created for the VQA task are used as contrast inputs for the localization task. For instance, in Fig. 3, the contrast outputs *frisbee* and *football* selected in Step 3 for the VQA task are used as localization queries (contrast inputs) in Step 4.

Finally, since image captioning is the task of generating a natural language description from an image, and text-to-image generation is the reverse process, one can reuse the ground truth annotations and contrasting annotations of captions for the task of image generation by simply reversing them. Similar to localization, the contrast outputs created for the image captioning task are used as contrast inputs for this task, and we measure the models' likelihood of generating the ground truth image in response to the contrast inputs.

**(Step 5) Manual Filtering.** This generated dataset was then subject to manual filtering and editing to ensure the high quality of the contrast sets. In this step, contrast sets that were synonyms, holonyms, hypernyms, or meronyms were removed from the dataset, in addition to other invalid perturbations. We conducted a study for inter-annotator agreement between two expert annotators on 200 samples and found

| Category | Input Image | Original Instances | Contrast Sets | V | G | L |
|---|---|---|---|---|---|---|
| Object | | **Caption**: A child in a bed with a striped sweater and colorful blanket. **VQA**: What is the baby sleeping with? blanket **Image Generation**: Generate an image with the text "A child in a bed with a striped sweater and colorful blanket." | stuffed animal | ✗ | ✗ | - |
| | | | pillow | ✓ | ✓ | - |
| | | | teddy bear | ✗ | ✓ | - |
| Attribute | | **Caption**: A brown and white cat lying on the bed. **VQA**: What color are the cat's spots? white **Image Generation**: Generate an image with the text "A brown and white cat lying on the bed." | yellow | ✗ | ✗ | - |
| | | | black | ✗ | ✗ | - |
| | | | gray | ✗ | ✗ | - |
| | | | orange | ✓ | ✗ | - |
| Food | | **Caption**: Apples hanging from a tree that has hardly any leaves on it. **VQA**: What type of tree is this? apple **Image Generation**: Generate an image with the text "Apples hanging from a tree that has hardly any leaves on it." **Localization**: Which region does the text "apple" describe? | pear | ✗ | ✓ | ✓ |
| | | | olive | ✓ | ✗ | ✗ |
| | | | orange | ✗ | ✗ | ✗ |
| | | | cantaloupe | ✓ | ✓ | ✓ |
| Animal | | **Caption**: Office space with cat on the television and work. **VQA**: What is playing on the TV? cat **Image Generation**: Generate an image with the text "Office space with cat on the television and work." **Localization**: Which region does the text "cat" describe? | cartoon | ✗ | ✗ | ✗ |
| | | | squirrel | ✗ | ✗ | ✗ |
| | | | baseball | ✗ | ✓ | ✗ |
| | | | butterfly | ✗ | ✗ | ✗ |
| | | | bowling | ✓ | ✓ | ✗ |
| Location | | **Caption**: A woman eating a piece of pizza near the kitchen counter. **VQA**: Where is the person standing? Kitchen **Image Generation**: Generate an image with the text "A woman eating a piece of pizza near the kitchen counter." | living room | ✗ | ✗ | - |
| | | | dining room | ✓ | ✗ | - |
| Role | | **Caption**: A man holding out to catch a baseball. **VQA**: What sport is the man playing? baseball **Image Generation**: Generate an image with the text "A man holding out to catch a baseball." | tee ball | ✓ | ✓ | - |
| | | | softball | ✗ | ✗ | - |
| | | | football | ✓ | ✗ | - |
| | | | basketball | ✓ | ✗ | - |
| Action | | **Caption**: A herd of zebra standing on top of a dry grass field. **VQA**: What are the two zebra doing on the left? standing **Image Generation**: Generate an image with the text "A herd of zebra standing on top of a dry grass field." | running | ✗ | ✗ | - |
| Person | | **Caption**: A young boy getting ready to blow out candles for his birthday. **VQA**: Is this child a boy or girl? boy **Image Generation**: Generate an image with the text "A young boy getting ready to blow out candles for his birthday." | girl | ✗ | ✗ | - |
| OCR | | **Caption**: A giant colgate clock sits on the shore next to water. **VQA**: What is the sponsor named? colgate **Image Generation**: Generate an image with the text "A giant colgate clock sits on the shore next to water." | walgreens | ✓ | ✗ | - |
| | | | billboard | ✗ | ✗ | - |
| | | | state farm | ✓ | ✗ | - |
| | | | wrigley's | ✓ | ✗ | - |
| | | | yves saint lauren | ✓ | ✗ | - |
| Misc. | | **Caption**: There are 4 sheep grazing in a field. **VQA**: How many sheep can be seen? 4 **Image Generation**: Generate an image with the text "There are 4 sheep grazing in a field." | 3 | ✓ | ✓ | - |
| | | | 5 | ✓ | ✓ | - |
| | | | 6 | ✓ | ✓ | - |
| | | | 2 | ✓ | ✓ | - |
| | | | 7 | ✓ | ✓ | - |

Figure 4: **Examples of contrast sets used in CocoCon.** For each example, we show the relevant image (left), the ground truth caption, VQA question, or image generation prompt for the image with the perturbed concept in green (middle), the set of perturbations used to generate alternative answers and predictions from Unified-IO $_{XL}$ for VQA (V), image generation (G) and localization (L) (right columns). ✓ and ✗ indicate scenarios where the model predictions for captioning and the corresponding task for that particular contrast set are consistent and inconsistent respectively. '-' denotes a lack of localization annotations for the sample.

an agreement for 98% of the data, indicating the high quality of the dataset. We prioritized the collection of clean, expert annotations over size for this probing dataset. Note that the contrast sets were manually filtered to ensure high quality at test, but at training time we only use automatically generated data.

**Note**: Our contrast set generation pipeline precedes recent advances in multimodal large language models (Liu et al., 2023a;b) that can be used to significantly ease the generation of contrast sets (see Appendix B.1).

## 4.2 Dataset Categories & Statistics

Each sample in the CocoCon dataset contains a set of ground truth annotations and a semantic concept within the original caption is replaced with multiple contrast sets. The ground truth annotations comprise

those for image captioning, VQA, and text-to-image generation, and 30% of the CocoCon samples also contain annotations for localization.[1] In total, the CocoCon dataset contains 4789 contrast sets for 1,500 samples from the COCO validation split, with an average of 3.2 contrast sets per sample. The semantic concepts used for perturbing the pivot instances in this dataset range from a large variety of semantic, syntactic, and grounding phenomena. We labeled each sample from CocoCon for these phenomena, see examples in Fig. 4 and a breakdown of the categories as well as additional examples in Appendix B. Attributes (color, height, material, etc.), inanimate objects, and food are the most frequent semantic concept categories in CocoCon, followed by animals, roles, actions, and location.

## 4.3 Evaluation

We measure the consistency between the captioning task (anchor) and each of the evaluation tasks independently. To evaluate consistency between captioning and VQA tasks, we compare the models' likelihoods of generating the caption and the VQA answer for both, pivot and contrast instances. For the localization and text-to-image generation tasks, the outputs are common to both, pivot and contrast instances, whereas the inputs contain semantic perturbations (see Fig. 3, Step 4). Hence, we compare the models' likelihood of generating the output in response to the input from the pivot instance $(x_t, y_t)$ vs. the input from the contrast instance $(\tilde{x}_t, y_t)$ i.e., we replace $f_\theta(\tilde{y}_{t_0}|x_{t_0}), f_\theta(\tilde{y}_{t_1}|x_{t_1})$ in Eqn. 1 with $f_\theta(y_{t_0}|\tilde{x}_{t_0}), f_\theta(y_{t_1}|\tilde{x}_{t_1})$ respectively. For example, we compare models' likelihood of generating the ground truth image in response to the gold caption and the contrast caption (e.g. caption containing *frisbee* vs. *football* in Fig. 3) for the text-to-image generation task.

# 5 Consistency-based Training

---

**Algorithm 1** Cross-Task Consistency-based Training

1: $\gamma \leftarrow$ ratio of consistency-based updates to total updates
2: $\lambda \leftarrow$ weight co-efficient for consistency-based loss
3: $t_0, [t_1, t_2, t_3] \leftarrow$ anchor task (e.g. captioning), evaluation tasks
4: $(x_{t_i}, y_{t_i}, \{\tilde{y}_{t_i}\}) \leftarrow$ input, gold output, contrast outputs for $t_i$
5: **for** $epoch = 1, 2, \ldots, N$ **do**
6:    **for** $step = 1, 2, \ldots, M$ **do**
7:        $r \leftarrow$ random(0, 1)
8:        **if** $r \leq \gamma$ **then**
9:            $i \leftarrow$ random(1, 2, 3)
10:           Anchor task: $(X_{t_0}, Y_{t_0}, \{\tilde{Y}_{t_0}\}) \leftarrow (x_{t_0}, y_{t_0}, \{\tilde{y}_{t_0}\})$
11:           Evaluation task: $(X_{t_i}, Y_{t_i}, \{\tilde{Y}_{t_i}\}) \leftarrow (x_{t_i}, y_{t_i}, \{\tilde{y}_{t_i}\})$
12:           Cross-entropy losses: $\{L_{ce}^0\}, \{L_{ce}^i\}$
13:           Ranks: $R_0, R_i \leftarrow$ rank$(\{L_{ce}^0\})$, rank$(\{L_{ce}^i\})$
14:           $L_{const} \leftarrow$ spearmanr$(R_0, R_i)$
15:           $L \leftarrow \lambda * L_{const} + L_{ce}$
16:       **else**
17:           Standard pretraining data: $(X, Y) \leftarrow \{x, y\}$
18:           Cross-entropy loss: $\{L_{ce}\}$
19:       **end if**
20:       Compute backward pass
21:   **end for**
22:   Evaluate updated model for cross-task consistency
23: **end for**

---

A unified model exhibiting inconsistent predictions suggests that the model has learned weak semantic representations that are sensitive to task variations. It is undesirable to work with a model that is susceptible to such frailties. Moreover, consistency constraints can provide useful information for learning well-rounded semantic representations (Lu et al., 2021) and reduce the need for training data (Zamir et al., 2020). Hence, we propose to train unified models in a way that preserves consistency across their predictions (see Algorithm 1). Given a pair of train instances $x_{t_0}, x_{t_1}$ for the tasks $t_0, t_1$, let $\{y_{t_0}\}, \{y_{t_1}\}$ be the spaces of $K$ probable and semantically equivalent outputs. $f_\theta(.)$ is the scoring function for a model with parameters $\theta$ and $\mathcal{R}(.)$ is some ranking function. Since ranking is a non-differentiable operation, we use soft ranking via regularization (Blondel et al., 2020) as the differentiable ranking function $\mathcal{R}(.)$. We formulate the consistency-based loss objective using Spearman's correlation as follows:

$$\mathcal{L}_{const} = \frac{1}{2}||\mathcal{R}(f_\theta(\{y_{t_0}\})) - \mathcal{R}(f_\theta(\{y_{t_1}\}))||^2 \tag{3}$$

Within a space of $k$ probable outputs for either task, if an output for task $t_0$ is ranked at $k - 2$ while the equivalent output for task $t_1$ is ranked at $k + 2$, the gradients from this objective are designed to push the two misaligned outputs towards a common rank $k$, which increases consistency as per the definition of $\mathcal{C}_k$ in Sec. 3. This can affect the task-specific accuracy of an inconsistent model, especially when the more probable

---

[1]Localization annotations are present when a COCO object appears in the gold caption and VQA answer.

Table 1: Summary of the CocoCon tasks supported by the various models used in our experiments.

| Model | Image Captioning | Visual QA (VQA) | Localization | Text-to-Image Gen. |
|---|:---:|:---:|:---:|:---:|
| Unified-IO (Lu et al., 2022) | ✓ | ✓ | ✓ | ✓ |
| OFA (Wang et al., 2022) | ✓ | ✓ | ✓ | finetune |
| Kosmos-2 (Peng et al., 2024) | zero-shot | zero-shot | ✓ | ✗ |
| GILL (Koh et al., 2023) | zero-shot | zero-shot | ✗ | ✓ |

output is the gold label. Hence, we minimize our proposed consistency objective in addition to the standard cross-entropy loss during training i.e.

$$\mathcal{L} = \lambda * \mathcal{L}_{const} + \mathcal{L}_{ce} \tag{4}$$

where $\mathcal{L}_{ce}$ is the cross-entropy loss and $\lambda$ is the weighting factor for the consistency objective. See Algorithm 1.

# 6 Experimental Setup

**Vision-Language Models.** Unified-IO (Lu et al., 2022) and OFA (Wang et al., 2022) are two recent publicly released models that perform a wide variety of tasks, including all tasks supported in the CocoCon benchmark. Unified-IO is pre-trained on all tasks in CocoCon, as well as multiple other vision-only, language-only and vision-language tasks. OFA models are pretrained on image captioning, VQA, image-infilling, and language-only tasks. Hence, we finetune the pretrained OFA models on the tasks supported in CocoCon for two epochs to support text-to-image generation.[2] We evaluate all size variations of both models. Additionally, we evaluate Kosmos-2 (Peng et al., 2024) and GILL (Koh et al., 2023) which support localization and text-to-image generation tasks respectively. Besides, both models support *zero-shot* image captioning and VQA tasks. See a summary of these models' capabilities in Tab. 1.

**Evaluation Metrics.** As outlined in Sec. 3, we compute consistency % ($\mathcal{C}_k$) and `spearmanr` ($\rho_{rank}$) for evaluating cross-task consistency. Additionally, we measure the following task-specific metrics: CIDEr score (Vedantam et al., 2015) for image captioning, accuracy for VQA (Goyal et al., 2017), IOU score (Padilla et al., 2020) for localization, and FID score (Heusel et al., 2017) for text-to-image generation.

**Consistency-based Training.** We begin with the finetuned checkpoint for the OFA$_{LARGE}$ model and continue training with the objective proposed in Sec. 5. We adapt the automated pipeline introduced in Sec. 4 to generate nearly 84K contrast sets from the training split of COCO Captioning, VQA, and localization datasets. We performed a manual analysis of this dataset and found that nearly 85% of the contrast sets are valid, which is of sufficient quality for large-scale training purposes. We use the cross-entropy loss as the score $f_\theta(.)$ function for each sample. The models are subjected to continued pretraining for one epoch and trained on the combination of contrast sets and original datasets for the four tasks in CocoCon. We set $\lambda = 0.25$ and use a learning rate of 1e-6. Additional hyperparameters can be found in Appendix C. This finetuned model is referred to as OFA$_{CON}$ in the rest of the paper.

# 7 Results

In this section, we discuss our findings from the evaluation of pretrained vision-language models, calibration of model likelihoods, common failure modes across models, and consistency-based training (see Sec. 5).

## 7.1 Evaluation of Pretrained Models

**Models are more inconsistent across tasks of diverse modalities.** We wish to study how the semantic understanding of a unified model varies with tasks. We evaluate the best (and largest) OFA and Unified-IO models on CocoCon and compare % consistency across the 3 tasks i.e., VQA, localization, and text-to-image generation, with respect to the anchor task, i.e. image captioning. Results are shown in Fig. 5(a). For VQA (blue plots), OFA $_{HUGE}$ and Unified-IO $_{XL}$ models exhibit 78% and 68% top-1 consistency respectively.

---

[2]The FID score of our finetuned OFA models on the text-to-image generation task is higher (worse performance) than that reported in Wang et al. (2022) because the latter model is finetuned on the text-to-image generation task only.

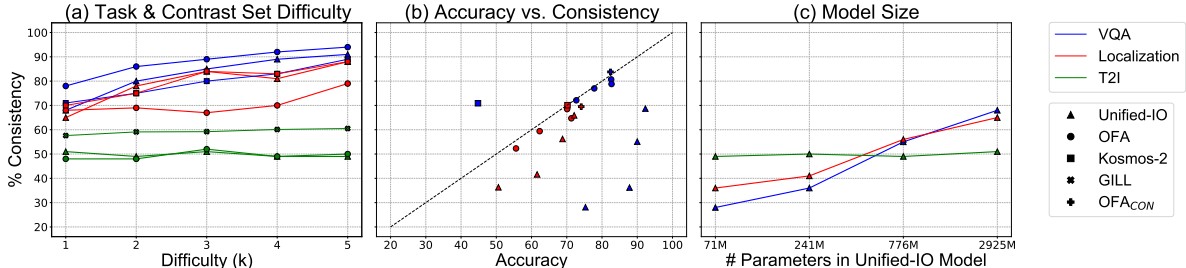

Figure 5: **Results from evaluation on the CocoCon benchmark.** (a) % Consistency of Unified-IO $_{XL}$, OFA$_{HUGE}$, Kosmos-2 and GILL models for varying difficulty ($k$) and all tasks in CocoCon, (b) comparison of % accuracy with % consistency ($k$=1) values for all models evaluated in this paper and our OFA$_{Con}$ model (see Sec. 5), and (c) % consistency ($k$=1) values for different sizes of Unified-IO models.

This number changes to 68% and 65% top-1 consistencies for localization (red plots), respectively, suggesting that unified models are especially prone to variation in semantic understanding when the outputs belong to different modalities. This is further supported by results for image generation (green plots) with 48% and 50% top-1 consistencies. Text-to-image generation is more complex than localization, because of the high dimensional output and rigorous semantic understanding required for the task. These results also suggest that cross-task inconsistency increases with the complexity of the task as well.

**Models are inconsistent at hard as well as easy contrast sets.** The contrast sets used for evaluating top-1 % consistency are *hard negatives* and we observe low consistency for these samples (see Fig. 5(a)). For easier contrast sets i.e. in $k > 1$ scenarios, the % consistency increases significantly (yet remains $< 100\%$) for tasks with outputs of the same modality as the anchor task, as seen for VQA in Fig. 5(a). The % consistency for VQA increases from $k$=1 to $k$=2 by 12%, 8%, and 4% for Unified-IO, OFA, and Kosmos-2 respectively, and then by smaller margins with increasing k. Similarly, the % consistency for localization increases from $k$=1 to $k$=2 by 12%, 1%, and 5% for Unified-IO, OFA, and Kosmos-2 respectively, and then by smaller (or negative) margins. However, we see almost no improvement in consistency for text-to-image generation with increasing $k$, implying that the unification of modalities within a model is a non-trivial challenge.

**Models are more accurate than consistent.** We compare the top-1 % consistency scores with the task-specific accuracies of models on the CocoCon dataset in Fig. 5(b), and observe that consistency and accuracy are positively correlated. Unified-IO models feature well below the $x = y$ line while other models, i.e., OFA and Kosmos-2, lie close to the $x = y$ line. Unified-IO models are more accurate at the VQA and the localization tasks than the other models, without demonstrating a similar gain in consistency. This suggests that when the Unified-IO models make mistakes for one task, they rarely make the same kind of mistakes on the other tasks, which is what would allow a model to achieve high consistency independently of accuracy. On the other hand, OFA and Kosmos-2 models show a tight correlation between accuracy and consistency. Additionally, we observe that the models gradually transition over the $x = y$ line with increasing $k$ i.e., easier contrast sets (see Fig. 13). Notably, OFA$_{CON}$ consistently ranks the highest in terms of consistency at all $k$. Ideally, we want models to be highly consistent across tasks despite being inaccurate i.e., lie well above the x=y line, and theoretically, it is possible with a unified semantic backbone in the model (see discussion in Appendix A). Instead, existing models appear to be consistent mostly by virtue of being accurate. This has the worrying implication that harder or more ambiguous tasks will lead to severe inconsistencies, and high consistency on easy tasks does not necessarily mean models are parsing inputs in a unified way across tasks. It also highlights the importance of studying hard tasks like image generation when evaluating consistency.

**Models capable of performing more tasks are more inconsistent.** Unified-IO models are trained on 90 diverse datasets from vision and language domains and can perform all 7 tasks on the GRIT benchmark (Gupta et al., 2022a). In contrast, OFA models are pretrained on a subset of the tasks that can be performed by Unified-IO. Interestingly, we observe that OFA models are more consistent than Unified-IO across all three tasks in the CocoCon benchmark. Additionally, Kosmos-2 and GILL (see rows I,J in Tab. 2) are more consistent than any Unified-IO or OFA models at their specialized tasks i.e., localization and text-to-image

Table 2: **Results from evaluation on the CocoCon benchmark.** Metrics are task-specific accuracies, % consistency ($k = 1$) and Spearman's rank correlation coefficient ($\rho_{rank}$). Higher is better, except for FID.

| | Model | Param | Caption CIDEr | VQA Acc. | $\mathcal{C}_1$ | $\rho_{rank}$ | Localization Acc. | $\mathcal{C}_1$ | $\rho_{rank}$ | Text-to-Image Gen. FID ↓ | $\mathcal{C}_1$ | $\rho_{rank}$ |
|---|---|---|---|---|---|---|---|---|---|---|---|---|
| **A** | Unified-IO$_{Small}$ | 71M | 111.8 | 75.3 | 28.1 | -0.06 | 50.6 | 36.3 | -0.09 | 93.45 | 49.5 | 0.05 |
| **B** | Unified-IO$_{Base}$ | 241M | 140.5 | 87.8 | 36.2 | 0.12 | 61.59 | 41.6 | 0.13 | 91.56 | 50.4 | 0.02 |
| **C** | Unified-IO$_{Large}$ | 776M | 227.1 | 90.0 | 55.1 | 0.36 | 68.8 | 56.2 | 0.03 | 85.04 | 48.5 | -0.01 |
| **D** | Unified-IO$_{XL}$ | 2.9B | **269.9** | **92.3** | 68.7 | 0.48 | 72.1 | 65.9 | 0.20 | 70.23 | 50.8 | -0.0 |
| **E** | OFA$_{Medium}$ | 93M | 83.4 | 72.7 | 72.1 | **0.67** | 55.6 | 52.3 | 0.21 | 110.3 | 49.1 | -0.02 |
| **F** | OFA$_{Base}$ | 182M | 100.7 | 77.8 | 77.0 | 0.65 | 62.3 | 59.4 | 0.19 | 105.7 | 50.1 | 0.04 |
| **G** | OFA$_{Large}$ | 472M | 113.5 | 82.6 | **80.7** | 0.64 | **71.3** | 64.7 | 0.28 | 103.4 | 52.3 | 0.02 |
| **H** | OFA$_{Huge}$ | 930M | 110.3 | 82.7 | 78.8 | 0.62 | 70.1 | 68.5 | 0.33 | 107.3 | 48.3 | -0.01 |
| **I** | Kosmos-2 | 1.6B | 65.8 | 44.8 | 70.9 | 0.62 | 70.8 | **70.1** | **0.60** | - | - | - |
| **J** | GILL | 8B | 45.6 | 35.7 | 51.9 | 0.41 | - | - | - | **25.4** | **57.6** | **0.10** |
| | | | Consistency-based Training | | | | | | | | | |
| **G** | OFA$_{Large}$ | 472M | 113.5 | 82.6 | 80.7 | 0.64 | 71.3 | 64.7 | 0.28 | 103.4 | 52.3 | 0.02 |
| **K** | + Cont. Pretrain | 472M | 118.8 | 82.7 | 81.1 | 0.63 | 73.5 | 65.9 | 0.27 | **98.5** | 51.7 | 0.04 |
| **L** | + Hinge Loss | 472M | 117.5 | **83.0** | 82.9 | 0.64 | 73.8 | 67.7 | 0.29 | 99.5 | 53.2 | 0.05 |
| **M** | OFA$_{CON}$ (ours) | 472M | 119.4 | 82.4 | **83.8** | **0.67** | **74.1** | **69.5** | **0.35** | 99.1 | **53.8** | **0.09** |

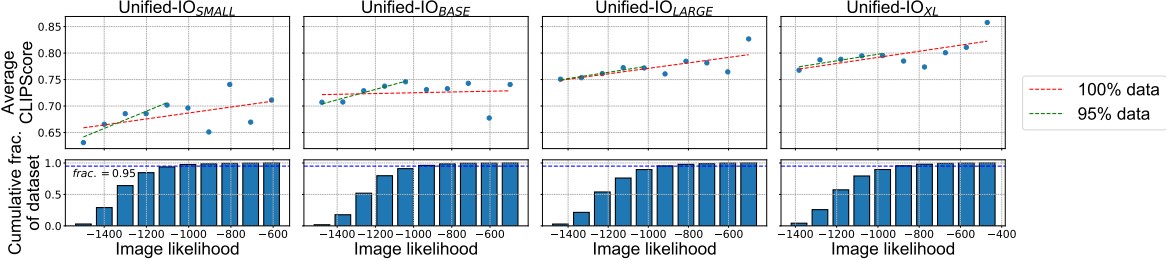

Reliability maps of model calibration for the text-to-image generation task

Figure 6: **Reliability maps (Guo et al., 2017) for likelihood scores from Unified-IO.** A positive correlation between likelihoods and output accuracy i.e. CLIPScore (Hessel et al., 2021) indicates that the likelihood scores can be reliable stand-ins for model confidence in our evaluation of the CocoCon benchmark.

generation respectively. This suggests that massive multi-tasking can lead to larger misalignment between models' semantic understanding across tasks, especially those with heterogeneous output modalities.

**Larger multi-task models that are more accurate are more consistent.** We evaluate various sized Unified-IO and OFA models (see Fig. 5(c) and Tab. 2). We see that the top-1 % consistency values increase generously with the scale of the model for VQA and localization, up to 20% increase from Unified-IO$_{SMALL}$ to Unified-IO$_{XL}$ on VQA. Improvements are modest for image generation with model size. We see similar trends in OFA models barring a small drop in accuracy as well as consistency in the largest model.

## 7.2 Calibration of Model Likelihoods for Text-to-image Generation

In our evaluation of multimodal models on the CocoCon benchmark, we compare the likelihood of a model for predicting one output vs. another and observe meaningful, intuitive trends for the VQA and localization tasks. However, the results are less sensitive to variations in model size and contrast set difficulty for the text-to-image generation task. Hence, in this section, we examine the calibration of the likelihoods obtained from Unified-IO and OFA models for text-to-image generation outputs, to find out whether they can be used reliably for evaluation on the CocoCon benchmark. Guo et al. (2017) state that the class probabilities predicted by a well-calibrated classifier should reflect the ground truth correctness likelihood of the model. They demonstrate this behavior through reliability maps, where first, the range of class probabilities predicted by a model is divided into equally spaced bins. In each bin, the average predicted probability of the samples belonging to the bin is compared to the average accuracy of the samples. This plot should align with the $x = y$ line for a well-calibrated model, indicating that the model confidence is an exact estimate of how

likely it is for the prediction to be correct. We adapt this principle for the text-to-image generation task in multimodal models and plot reliability maps for all model sizes of Unified-IO by comparing the likelihood of predicted images with their average CLIPScore (Hessel et al., 2021). See results in Fig. 6.

The ideal scenario of $x = y$ for classification tasks is ill-defined for the text-to-image generation task since likelihood scores do not have a finite lower bound. Hence, we only seek a positive correlation between the likelihood scores and CLIPScore values as an indication that the model is well-calibrated. We find that Unified-IO models' likelihoods are indeed positively correlated with CLIPScore values for all model sizes (see linear regression fit over data points in Fig. 6). The correlation sometimes falters towards the upper bound of the distribution which can be attributed to the small sample size in those bins, as seen in the distribution of the cumulative fraction of the dataset over the range of likelihood scores (see Fig. 6; bottom). A linear regression fit over 95% of the samples yields an even stronger positive correlation, especially for the smaller Unified-IO models. This analysis suggests that the low values of cross-task consistency as seen for the text-to-image generation task in Fig. 5 primarily stems from the complexity of the task itself as well as the gross misalignment between text and image output modalities in multimodal models. Further, we perform temperature calibration (Guo et al., 2017) on the likelihoods from Unified-IO$_{SMALL}$ and observe no significant improvement in correlation (see discussion and supporting figures in Appendix D.1). We also perform this analysis on likelihoods of text-to-image generation outputs from OFA models and find that they are similarly reliable for evaluation on the CocoCon benchmark.

### 7.3 Common Failure Modes

We analyze the contrast sets in CocoCon for which Unified-IO$_{XL}$, OFA$_{Huge}$, Kosmos-2 models are *inconsistent for all tasks* and categorize the errors into the tags defined in Tab. 3. We find that all three of the models perform worst at recognizing *attributes* correctly, i.e., 39.7%, 34.2%, 25.5% of errors from Unified-IO$_{XL}$, OFA$_{Huge}$, Kosmos-2 respectively pertain to attributes, which are significantly higher than the category's 20.9% distribution in the dataset. The other prevalent error categories are commensurate with the distribution in Coco-Con i.e., *object*, *food*, *animal*, and *location*. See examples of errors from Unified-IO$_{XL}$ in Fig. 4.

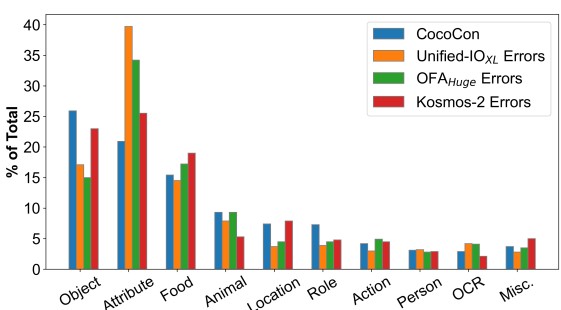

Figure 7: Comparison of categorical distribution in the CocoCon benchmark with that of errors from evaluation of Unified-IO$_{XL}$, OFA$_{Huge}$, Kosmos-2 models.

### 7.4 Consistency-based Training

As outlined in Sec. 5, we continue training OFA via the use of a cross-task consistency-based loss objective. Results for the finetuned model, OFA$_{Con}$, are shown in Tab. 2 (see rows K-M in Consistency-based Training). Since OFA$_{Con}$ (row M) is finetuned for an additional epoch, we also provide a baseline where OFA is finetuned for an additional epoch with just the original cross entropy objective (row K). We find that our proposed loss objective improves consistency along both metrics i.e. top-1 % consistency and rank correlation. The top-1 % consistency improves by 2% for VQA and text-to-image generation, and a larger margin i.e. 4%, for localization. Importantly, we see that this preserves the accuracy for VQA, tipping the model over the $x = y$ line in Fig. 2(b), improves localization by +0.6 points, and preserves the FID for text-to-image generation. These results show the benefits of incorporating consistency-based objectives while training GPV models.

## 8 Conclusion

We present a benchmark dataset, CocoCon, to probe cross-task inconsistency in unified multimodal models and a loss objective to improve the same. Our results demonstrate that cross-task inconsistency is a significant issue in such models and can be mitigated with our proposed loss objective. We hope that CocoCon serves as a useful resource for probing the reliability of unified multimodal models in the future.

## Limitations & Future Work

**Out-of-domain evaluation.** The CocoCon benchmark is the first step (to the best of our knowledge) towards measuring consistency across tasks in multimodal models. It is derived from the COCO dataset (Lin et al., 2014) which has been widely used for pretraining multimodal models, hence, the data samples in CocoCon are in-domain for these models. Despite CocoCon being sampled from the same distribution as the training data of such models, we are seeing significant inconsistencies across tasks, especially those of different output modalities. We expect that these inconsistencies will aggravate when the samples are drawn from an out-of-domain distribution. Our work does not conduct out-of-domain evaluation, nevertheless, our automated dataset creation pipeline and evaluation framework provide the groundwork for preparing benchmarks similar to CocoCon for out-of-domain-evaluation in future research.

**Additional tasks.** Our evaluation and analysis are carried out on three tasks that do not represent the full spectrum of tasks in the multimodal space. However, as shown in Fig. 2, this evaluation framework can be easily extended to more vision-and-language tasks. For instance, the consistency between image captioning and referring expression comprehension (Kazemzadeh et al., 2014) can be computed in the same way as done for localization. In this case, the semantic concepts appearing in the referring expression should appear in the caption, and one or more of those concepts should be replaced with contrast sets. The ground truth bounding box of the referring expression can be used to estimate likelihoods (as outlined for localization in Sec. 4.3). Similarly, to compute cross-task consistency between image captioning and visual entailment (Xie et al., 2019), the hypotheses that contain one or more semantic concepts overlapping with the caption should be perturbed with contrast sets. Further, our work capitalized on creating contrast sets based on text perturbations since they are easy to generate at scale and can be implemented for several vision-and-language tasks (Hu et al., 2022; Gupta et al., 2022b). However, contrast sets based on image perturbations can provide important, additional insights into cross-task consistency of unified vision-language models. Advances in text-guided image-editing (Zhang et al., 2023; Choi et al., 2023) can be leveraged to create such contrast sets at scale in future work.

**Availability of cross-task annotations.** We chose the COCO dataset as a starting point for our benchmark because it is a popular dataset that has been widely used for the training and evaluation of vision-language models. The availability of annotations for multiple tasks served as a convenient starting point as it allowed us to create annotations for multiple tasks with the same pivot instance. However, cross-task consistency is defined for a *pair of tasks* and is evaluated independently of the other tasks (see Sec. 3). Hence, to evaluate cross-task consistency, we do not require multiple task annotations for the same pivot instance. Instead, we can have different sets of pivot instances with annotations for a different pair of tasks, which can be easily scaled to multiple tasks. Further, many contemporary models super-specialize in one or more tasks, such as Segment Anything for image segmentation (Kirillov et al., 2023), Kosmos-2 for localization (Peng et al., 2024), BLIP-2 for captioning and VQA (Li et al., 2023). These models can be used to rapidly create annotations for a pair of tasks, and coupled with manual filtering, can be used for scaling to other datasets as well as tasks.

**Aggregation of likelihood over semantically similar outputs.** Ideally, when computing likelihood for a textual output such as the caption *A woman playing with her hair while sitting on a bed*, we should consider likelihoods for semantically similar captions as well e.g., those with invariant edits like *her hair→hair* and *a bed→bed*, paraphrases etc. This issue of considering highly similar outputs for likelihood estimation is a subset of a broader, open challenge of aggregating over semantically equivalent outputs during the evaluation of free-form natural language generations. We were mindful of this issue when creating the CocoCon benchmark. To maintain uniformity in the evaluation setting, all contrast sets in CocoCon are made of noun, verb, adjective, or preposition words only. Additional pronouns like *her* in *her hair* are neither replaced in the original caption nor added in the contrast sets if they are not significant semantic concepts. Some recent works propose methods for query or output expansion via mining, paraphrasing (Kuhn et al., 2022), multiple generations (Jiang et al., 2020), etc., and then ensemble over a pool of similar outputs to get an aggregate score estimate. Since this is an active area of research, we leave their integration with cross-task consistency evaluation to future work.

## Broader Impact Statement

The CocoCon benchmark is designed to test the cross-task consistency of unified multimodal models. Our evaluation exposes inconsistencies in such models, indicating that the model outputs are not sufficiently reliable for real-world deployment. We anticipate that our work will influence further research on the important topic of stress testing of unified vision-language models in the community.

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

## Overview

The Appendix is organized as follows:
**Section A**: Discussion on the relationship between cross-task consistency and task-specific accuracies, including results from simulation experiments.
**Section B**: Definitions of the categories in CocoCon, additional examples from CocoCon and examples of contrast sets generated using Multimodal Large Language Models (MLLMs).
**Section C**: Hyperparameters for training OFA$_{CON}$ model.
**Section D**: Additional results on model calibration, ablations from training of OFA$_{CON}$ and supporting figures.

## A    Relationship between Consistency and Accuracy

To understand the relationship between cross-task consistency and task-specific accuracies, we run a simulation experiment where we generate predictions from a hypothetical model for an anchor task and a target task under three different scenarios where the model makes: (A) independent errors on both tasks, (B) same errors on both tasks and (C) different errors on both tasks. See results in Fig. 8 and a discussion below.

When a model has 100% accuracy on the anchor task (see Sec. 3), the cross-task consistency of the model (with respect to the anchor task and a target task) is equal to the model's accuracy on the target task, irrespective of the model's semantic alignment across tasks (see Fig. 8). However, when anchor task accuracy is less than 100%, the outcome of a model's cross-task consistency ($C_1$) is closely tied with the nature of errors made by the model across tasks. If the model makes independent errors on anchor and target task, the cross-task consistency tends to stay close to 50% (see Fig. 8 A). If the model makes the same errors in both tasks, then the model's consistency generally remains high (see Fig. 8 B) and it features above the $x = y$ line in Fig. 5(b). If the model makes different errors for both tasks, the model's consistency remains low (see Fig. 8 C) and it features below the $x = y$ line in Fig. 5(b). The exceptions to these scenarios are when the anchor and target accuracies are either 0% or 100%. In such cases, the cross-task consistency is 100% if the accuracies of both tasks are the same, and 0% otherwise.

## B    The CocoCon Benchmark

See a selection of additional examples from the CocoCon benchmark in Fig. 9 and an explanation as well as a breakdown of the various contrast set categories in CocoCon in Tab. 3.

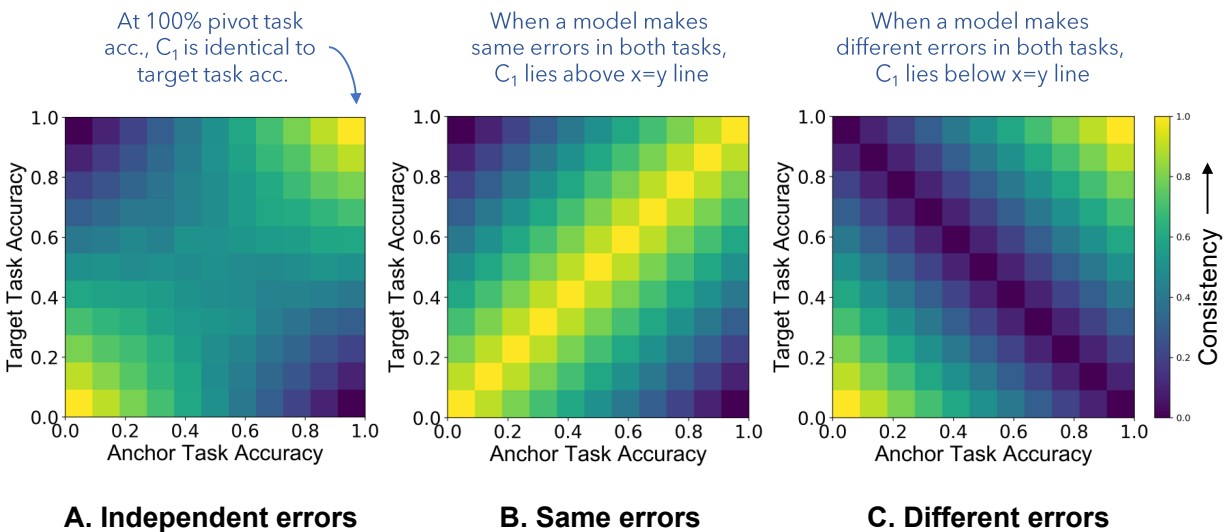

Figure 8: **Simulation of the relationship between pivot task accuracy and target task accuracy** under the scenarios where a model makes (a) independent errors in both tasks, (b) same errors in both tasks, and (c) different errors in both tasks.

Table 3: Definition of CocoCon categories and dataset statistics.

| Category | Description | # Samples | # Unique contrast sets |
|---|---|---|---|
| Object | All inanimate objects excluding food items. | 388 | 648 |
| Attribute | Adjectives used as modifiers to a noun e.g., color (*red* chair), height (*tall* building), size (*small*), material (*tiled* wall), etc. | 314 | 221 |
| Food | Food items including fruits, vegetables, and other cooked items. | 231 | 409 |
| Animal | Includes all mentions of animals, predominantly those featured in COCO objects. | 139 | 177 |
| Location | Includes broadly defined areas (e.g., *bathroom*, *hotel*, *library*), finer visual elements (e.g., *floor*, *sidewalk*), and spatial references (e.g., *inside*, *outside*, *on table*). | 111 | 143 |
| Role | Includes professional roles such as *chef*, *baseball player*, etc. | 109 | 74 |
| Action | Comprises transitive (e.g. *flying kite*) as well as intransitive actions (e.g. *sitting*, *standing*) performed by persons and animals. | 63 | 117 |
| Person | Concepts from one of the following: *man/male/guy*, *woman/female/lady*, *boy*, *girl*. | 47 | 16 |
| OCR | Texts present in the image e.g., writing on a cake, numbers on a digital clock, billboard, etc. | 43 | 132 |
| Misc. | All other minor sub-categories e.g., weather, direction, etc. | 55 | 116 |
| Overall | - | 1500 | 1820 |

## B.1 Generating Contrast Sets using Multimodal Large Language Models

As we note in Sec. 4, our work on the automated pipeline for generating contrast sets at scale precedes recent advances in open-source multimodal large language models (MLLMs) like LLaVA1.5 (Liu et al., 2023a). Fortunately, the multi-step process outlined in Sec. 4 can now be replaced with a single inference step using a state-of-art MLLM as we show below. To compare the diversity of contrast sets generated using our method to those generated using MLLMs, we perform a small experiment using LLaVA 1.5 (Liu et al., 2023a) where we generate contrast set candidates for the examples demonstrated in Fig. 4 by prompting the model to replace semantic concepts within captions directly. We use the following prompt and provide the corresponding image as input:

| Category | Input Image | Original Instances | Contrast Sets |
|---|---|---|---|
| Location |  | **Caption**: A surfer is surfing the waves in the ocean. 
 **VQA**: Where is the man surfing? ocean 
 **Image Generation**: Generate an image with the text "A surfer is surfing the waves in the ocean." | lake 
 river |
| Role |  | **Caption**: A gorgeous young lady holding a tennis racquet on a court. 
 **VQA**: What sport is this? tennis 
 **Image Generation**: Generate an image with the text "A gorgeous young lady holding a tennis racquet on a court. | badminton 
 ping-pong 
 polo |
| Person |  | **Caption**: A man is flying a kite on the beach. 
 **VQA**: Who is flying the kites? man 
 **Image Generation**: Generate an image with the text "A man is flying a kite on the beach." | woman 
 girl 
 boy 
 couple 
 dog |
| Attribute |  | **Caption**: A large white kitchen range with gold towels on the front. 
 **VQA**: What color is the stove in this room? white 
 **Image Generation**: Generate an image with the text "A large white kitchen range with gold towels on the front." | silver 
 black 
 sliver and black 
 blue 
 tan |
| Object |  | **Caption**: A cup with three pairs of scissors sitting on a table. 
 **VQA**: What is in this cup? scissors 
 **Image Generation**: Generate an image with the text "A cup with three pairs of scissors sitting on a table." 
 **Localization**: Which region does the text "scissors" describe? | pliers 
 utensils 
 pens |
| Food |  | **Caption**: A person cutting up a soccer ball cake. 
 **VQA**: Which one has been nibbled on? soccer ball cake 
 **Image Generation**: Generate an image with the text "A person cutting up a soccer ball cake." 
 **Localization**: Which region does the text "soccer ball cake" describe? | chocolate cake |
| Action |  | **Caption**: A professional tennis player about to serve. 
 **VQA**: What is the man about to do? serve 
 **Image Generation**: Generate an image with the text "A professional tennis player about to serve." | hit ball 
 swing 
 return ball 
 throw ball |
| Animal |  | **Caption**: Three elephants walking along a river near a jungle. 
 **VQA**: What animals are they? elephants 
 **Image Generation**: Generate an image with the text "Three elephants walking along a river near a jungle." 
 **Localization**: Which region does the text "elephant" describe? | lions 
 babies 
 apes |
| OCR |  | **Caption**: A yellow diamond shape sign indicating duck crossing. 
 **VQA**: What is the sign portraying? duck crossing 
 **Image Generation**: Generate an image with the text "A yellow diamond shape sign indicating duck crossing." | goose crossing 
 snake crossing |
| Misc. |  | **Caption**: A winter scene of a park bench covered in snow among the trees. 
 **VQA**: Is it summer or winter? winter 
 **Image Generation**: Generate an image with the text "A winter scene of a park bench covered in snow among the trees. | summer 
 fall 
 spring |

Figure 9: **Additional examples of contrast sets in CocoCon.** For each example, we show the relevant image (left), the ground truth caption, VQA question, or image generation prompt for the image with the perturbed concept in green (middle) and the set of perturbations used to generate alternative answers.

```
Based on the image, write 5 different answers for filling in the blank in the following
image caption:  [caption with blank placeholder ].  For each answer, write your
confidence in how well the answer fits into the caption while staying true to the image.
```

Table 4: **Examples of contrast sets generated using LLaVA1.5 (Liu et al., 2023a)**. Comparison of contrast sets generated using our method in Sec. 4 to those generated using a single-step inference from the LLaVA1.5 model. Concepts replaced in the caption are *emphasized*. The comparable diversity of both sets of contrast sets suggests that our multi-step pipeline can be significantly simplified with the use of MLLMs.

| Original Caption | CocoCon contrast sets (using VQA) | LLaVA 1.5 contrast sets (using caption) |
|---|---|---|
| A child in a bed with a striped sweater and a colorful *blanket*. | stuffed animal, pillow, teddy bear | stuffed animal, blanket, pillow, teddy bear |
| A brown and *white* cat lying on the bed. | yellow, black, grey, orange | black, orange, tan, cream |
| *Apples* are hanging from a tree that has hardly any leaves on it. | pear, olive, orange, cantaloupe | oranges, pears, peaches, plums |
| Office space with *cat* on the television and work. | cartoon, squirrel, baseball, butterfly, bowling | - |
| A man holding out to catch a *baseball*. | tee ball, basket ball, softball, football | frisbee, soccerball, tennis ball, golf ball |
| A giant *colgate* clock sits on the shore next to water. | walgreens, billboard, state farm, wrigleys, yves saint lauren | coca-cola, clocktower, clockface |

Table 5: Hyperparameters for training OFA$_{Con}$.

| Hyperparameter | Value |
|---|---|
| Proportion of ranking updates ($\gamma$) | 0.5 |
| Weight co-efficient of ranking loss ($\lambda$) | 0.25 |
| Regularization strength of soft ranking | 1.0 |
| Learning rate | 1e-6 |
| Max. train epochs | 1 |
| Batch Size | 2 |
| Warmup ratio | 0.1 |
| Label smoothing | 0.0 |

The results are presented in Tab. 4. Overall, we observe similar diversity in the contrast sets generated using VQA vs. image caption. This suggests that the multi-step contrast set generation pipeline in Sec. 4 can be significantly simplified with the use of MLLMs and suitable prompt engineering, and can be potentially used for extending the CocoCon benchmark to other vision-language tasks in a scalable manner.

## C  Training Hyperparameters

The complete hyperparameters for training OFA$_{Con}$ using the rank correlation-based loss objective are available in Tab. 5.

## D  Additional Results

### D.1  Calibration of Model Likelihoods for Text-to-image Generation

In Sec. 7.2, we note that we perform temperature calibration (Guo et al., 2017) of Unified-IO$_{SMALL}$ to examine whether it improves the calibration of the model likelihoods for outputs from the text-to-image generation task. See results in Fig. 10. We observe that the correlation does not improve significantly with different temperature values. We also compare the distribution of output likelihoods for various tasks and all model sizes of Unified-IO in Fig. 11. While we see different distributions for outputs from pivot instances and contrast instances for the VQA and localization tasks, we observe nearly complete overlap in the distributions of outputs for the text-to-image generation task, further supporting our conclusion that unified vision-language models do not have a unified semantic backbone across all supported tasks.

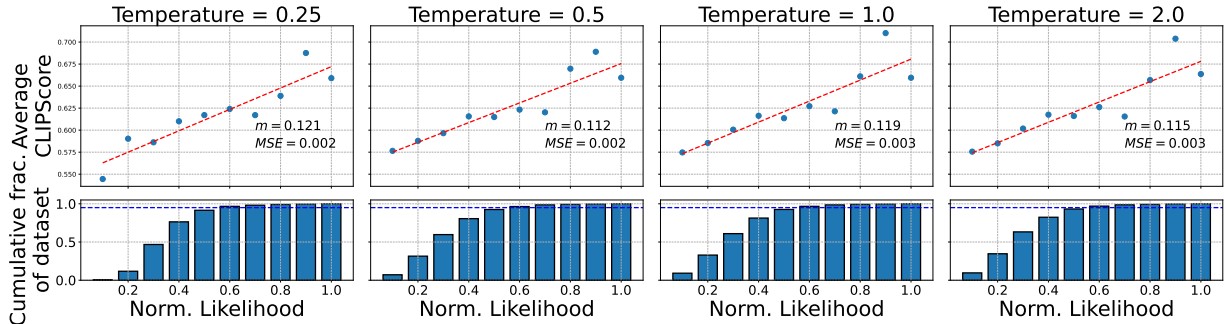

Temperature calibration of Unified-IO for the text-to-image generation task

Figure 10: **Temperature calibration (Guo et al., 2017) of the likelihood scores from Unified-IO**$_{SMALL}$ **model for the text-to-image generation task.** We measure the slope ($m$) and mean squared error (MSE) of the best linear regression fit to the temperature-calibrated likelihoods, and compare for different values of temperature ($t$). We do not observe significant differences in correlation strength ($m$) or linear fit (MSE) at values of $t$ other than 1, which is the default in our evaluation of the CocoCon benchmark.

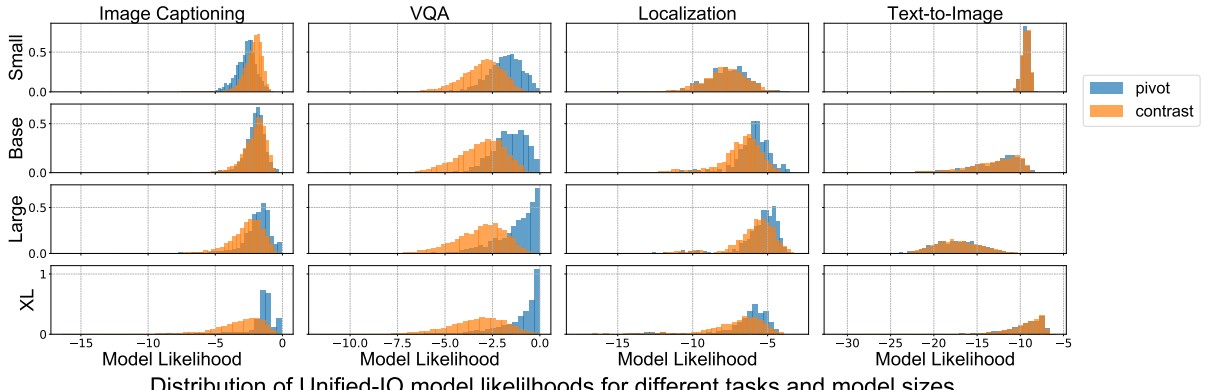

Distribution of Unified-IO model likelihoods for different tasks and model sizes

Figure 11: **Comparison of distribution of Unified-IO model likelihoods** for *pivot* and *contrast* instances, for various tasks in the CocoCon benchmark and all model sizes.

## D.2 Ablation Results & Examples

We present results from the ablation of the weight co-efficient ($\lambda$) hyperparameter for the consistency-based loss objective in Tab. 6. We observe that a higher $\lambda$ hurts accuracy while a lower $\lambda$ does not improve consistency. We also present examples where OFA$_{CON}$ is more consistent than the pretrained OFA$_{LARGE}$ in Fig. 12.

Table 6: **Results from ablation of the weight co-efficient ($\lambda$) for training of OFA$_{CON}$.** Metrics are task-specific accuracies, % consistency ($k = 1$) and Spearman's rank correlation coefficient ($\rho_{rank}$). Higher is better for all metrics except FID.

| Model | Params | Captioning CIDEr | VQA Acc. | $\mathcal{C}_1$ | $\rho_{rank}$ | Localization Acc. | $\mathcal{C}_1$ | $\rho_{rank}$ | Text-to-Image Gen. FID | $\mathcal{C}_1$ | $\rho_{rank}$ |
|---|---|---|---|---|---|---|---|---|---|---|---|
| Consistency-based Training | | | | | | | | | | | |
| OFA$_{CON}$ ($\lambda = 0.0$) | 472M | 118.8 | **82.7** | 81.1 | 0.63 | 73.5 | 65.9 | 0.27 | **98.5** | 51.7 | 0.04 |
| OFA$_{CON}$ ($\lambda = 0.25$) | 472M | **119.4** | 82.4 | 83.8 | 0.67 | **74.1** | 69.5 | 0.35 | 99.1 | 53.8 | **0.09** |
| OFA$_{CON}$ ($\lambda = 0.50$) | 472M | 117.8 | 81.8 | **84.2** | **0.70** | 73.1 | **69.9** | 0.35 | 99.3 | **54.1** | 0.08 |

| Category | Input Image | Original Instances | Contrast Sets | OFA$_{LARGE}$ V \| G \| L | OFA$_{CON}$ V \| G \| L |
|---|---|---|---|---|---|
| Location |  | **Caption**: A child in a bed with a striped sweater and colorful blanket. 
 **VQA**: What is the baby sleeping with? blanket 
 **Image Generation**: Generate an image with the text "A child in a bed with a striped sweater and colorful blanket." | stuffed animal 
 pillow 
 teddy bear | ✗ ✗ - 
 ✗ ✗ - 
 ✗ ✗ - | ✓ ✗ - 
 ✓ ✗ - 
 ✓ ✗ - |
| Animal |  | **Caption**: A white and green bus driving down a street. 
 **VQA**: What color is this school bus? green 
 **Image Generation**: Generate an image with the text "A white and green bus driving down a street. | blue 
 gray 
 tan | ✗ ✗ - 
 ✓ ✓ - 
 ✓ ✓ - | ✓ ✓ - 
 ✓ ✓ - 
 ✓ ✓ - |
| Action |  | **Caption**: A child sitting at a table putting a spoon in to a bowl. 
 **VQA**: What is the boy holding? spoon 
 **Image Generation**: Generate an image with the text "A child sitting at a table putting a spoon in to a bowl." 
 **Localization**: Which region does the text "spoon" describe? | spatula 
 fork 
 scoop 
 funnel 
 sponge | ✓ ✗ ✓ 
 ✓ ✗ ✓ 
 ✓ ✗ ✓ 
 ✓ ✗ ✓ 
 ✓ ✗ ✓ | ✓ ✗ ✓ 
 ✓ ✓ ✓ 
 ✓ ✗ ✓ 
 ✓ ✓ ✓ 
 ✓ ✗ ✓ |

Figure 12: **Examples from the CocoCon benchmark where OFA$_{CON}$ is more consistent than pretrained OFA$_{LARGE}$.** For each example, we show the relevant image (left), the ground truth caption, VQA question, or image generation prompt for the image with the perturbed concept in green (middle), the set of perturbations used to generate alternative answers and predictions from OFA$_{LARGE}$ and OFA$_{CON}$ for VQA (V), image generation (G) and localization (L) (right columns). ✓ and ✗ indicate scenarios where the model predictions for captioning and the corresponding task for that particular contrast set are consistent and inconsistent respectively. '-' denotes a lack of localization annotations for the given sample.

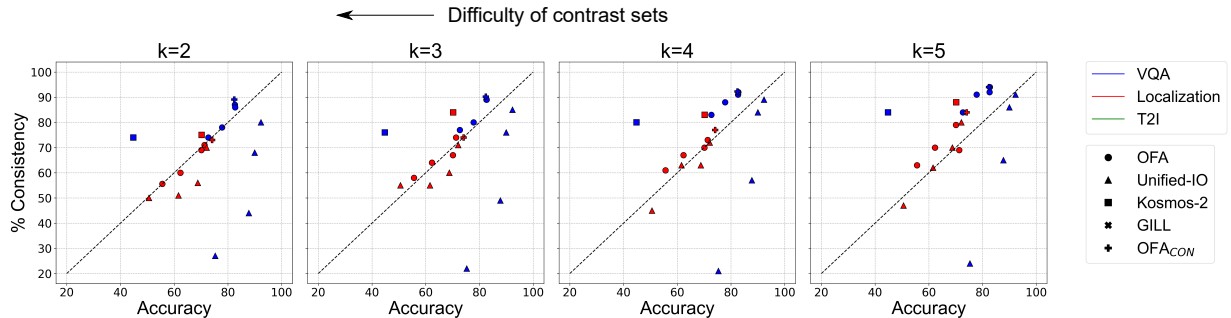

Figure 13: **Comparison of % accuracy with % consistency values** for all models evaluated in this paper and our OFA$_{Con}$ model (see Sec. 5) for decreasing difficulty of contrast sets ($k$=2,3,4,5). See Fig. 5(b) for performance on the hardest contrast sets i.e., $k = 1$.

## D.3 Consistency vs. Accuracy at Various Difficulty (k)

We compare the target task accuracies of all models evaluated in our experiments with the cross-task consistency at difficulty $k = 1$ ($C_1$) in Fig. 5(b) in the main text. To further support our analysis, we also present similar plots at decreasing difficulty levels i.e. $k = 2, 3, 4, 5$, in Fig. 13. With decreasing difficulty or for easier contrast sets, models gradually transition over the $x = y$ line which implies that models are generally more consistent than accurate at easier contrast sets.

