# OpenReview forum: "Exposing and Addressing Cross-Task Inconsistency in Unified Vision-Language Models"
_TMLR — Accepted by TMLR_

### Review · Reviewer_pVDY · 2023-09-29

**Summary Of Contributions:**

This paper proposes CocoCon, a benchmark dataset for measuring consistency of General Purpose Vision (GPV) model predictions across multiple related tasks. Example tasks include image captioning, visual question answering, localization, and text-to-image generation. CocoCon uses annotations from the validation split of the COCO dataset, and is constructed as follows (see Figure 3 for a nice overview):

1. The dataset is filtered to contain just “pivot instances” by comparing annotations for captioning and VQA. In this way it is ensured that the retained annotations concern the same concept, and thus predictions for them can be compared.

2.  Second, a number of “contrast sets” are created, which provide substitutes for a concept in the annotation. Suggestions for these are obtained from a pre-existing VQA model by considering its answer distribution for the remaining questions for each image.

3. A subsequent filtering step with a language model (T5) ensures that answers are valid completions to the question in terms of grammar, etc.

4. The detection labels for the localization tasks are filtered to retain only those that are present in the caption and corresponding VQA annotations. In this way, the concepts in the contrast set obtained in the previous step can be used as queries for the detection task, Here the models’ likelihood of generating the ground-truth bounding box in response to any of the queries is used for evaluation. Similarly for the generation task, the likelihood of generating the ground-truth image is measure for evaluation in response to an input (having substituted the concept with one from the contrast set).

5. A manual filtering steps was used to remove invalid concepts for contrasting, such as hypernyms. A study with two expert annotators on 200 samples revealed high consistency between their predictions. The manual filtering was only done for test sets, not for training.

The paper proceeds by evaluating two GPV models on CocoCon: Unified-IO and OFA. Two additional models are considered (Kosmos-2 and GILL), which only support a subset of the four tasks. Evaluation proceeds by comparing the consistency of the predictions of the model across tasks using substitutions from the contrast set. A model is deemed consistent if its preference for a particular label on the anchor task (here captioning) persists on the other tasks. Metrics include “% consistency @ k”, which is the fraction of the tasks the model is consistent for using the kth element of the contrast set (ordered by difficulty), and rank correlation over the ranked elements of the contrast set for two tasks. The latter matters global consistency of the output spaces.

Finally, a regularizer is proposed for training models to be more consistent. It works by using a soft ranker (that can be differentiated through) and a consistency-based loss that measures how well aligned the predictions are between two output spaces (tasks).

**Audience:**

Yes

**Broader Impact Concerns:**

no concerns.

**Claims And Evidence:**

Yes

**Requested Changes:**

I listed a number of questions regarding the analysis that I would like to see addressed in the previous section. Additionally, please consider the following questions for securing my recommendation:

* Could you please add additional examples of the dataset to the appendix, maybe 10 or 20 randomly selected or so? This would be helpful to review the quality of it.

* Please include a discussion of limitations of the proposed benchmark and analysis carried out.

* What is the process for releasing this dataset, and how will it be licensed?

**Strengths And Weaknesses:**

## Strengths

The paper poses an interesting research question, which is how well aligned are model predictions across output spaces in terms of the scores they assign to similar labels. This question is well-motivated (indeed human-computer interaction requires a certain degree of reliability) and increasingly more relevant as more general models are being developed that are capable of multiple different vision tasks. Furthermore:

* The paper is reasonably well written, and includes many visualizations, which are really helpful for giving an intuition of the dataset generation procedure and how evaluation works.

* The design of the dataset is quite innovative, leveraging pre-trained VQA models and language models to help generate contrast sets, while using human evaluating and filtering for quality control. The likelihood-based evaluating using the ground-truth image for t2i eval is quite clever, though it is not quite clear to me how well calibrated model likelihoods are for this task.

* The insight that the considered models are often not consistent is not very surprising, but it is useful to have some quantitative measure of this. There is a small increase in consistency as a function of contrast-difficulty (Figure 5a), which is encouraging to see. Figure 5c, which shows that VQA and Localization consistency (but not T2I performance) improves as a function of model scale is quite interesting as well.

* I certainly expect follow-up works to make use of this dataset.

## Weakenesses

* I don’t quite agree with some of the analysis presented in the paper. For example, it is written that “in k > 1 scenarios, the % consistency increases steeply” in response to Figure 5(a), though only a modest improvement of about 2-3% per step can be observed. Or “Most models feature below the x = y line ” for Figure 5b. While strictly speaking this is certainly true, barring Unified-IO it is more accurate to speak of most models falling “on” the line in my view, as the accuracy is quite a good predictor of consistency, barring a small intercept.

* The scatter plot in Figure 5b is only concerned with k=1, while I would like to see this analysis extended to greater values of k. It seems to be that this data is already available from Figure 5a and it is just a matter of plotting.

* Though the text-to-image setting is cleverly evaluated, it is questionable how well the models considered support this kind of evaluation. In particular, there is close to no sensitivity with regards to difficulty or model size in Figure 5, and the absolute consistency is very low. One explanation is that this task is the hardest and model struggle generally with this. Another, is that this kind of likelihood-based evaluation is ill-suited to evaluate model consistency. In particular, it is unclear to me how well calibrated likelihoods are for these models. Some follow-up analysis, such as looking at the distribution of likelihoods the model assigns in response to captions with substituted concepts could be informative.


## Questions

* What is the standard error of these models on this task? Is it safe to compare mean performance across samples or are there significant deviations between samples? Please update to plots to provide an indication of this (Figure 5a, 5c, 6)

* The effect of training with consistency based regularization is quite small, which is unexpected. What is the hypothesis for why this supervised training procedure does not fully close the gap?

---

> ### Author Response · Authors · 2023-10-25
> **Response to questions and requested changes**
>
> Dear Reviewer,
>
> We thank you for taking the time and effort to review our manuscript and provide detailed feedback! We appreciate that you have found our work to be interesting, useful, and well-written! We have addressed your questions and concerns in the revised manuscript (see blue text), outlined the main changes below, and pointed to relevant sections or updated figures wherever applicable. Please let us know if you have any additional concerns, we will be happy to address those!
>
> --------------------------------------------------------------------------------------------------------------------------------------------------------------
> * **Analysis of results in Section 7.1**:
>
> We agree that some of the statements in Section 7.1 need rephrasing and/or additional details to convey the varied results seen in the plots. We have rewritten the conclusions under the heading “Models are inconsistent at hard as well as easy contrast sets.” in Sec. 7.1 as follows:
>
> For easier contrast sets i.e. in k>1 scenarios, the % consistency increases significantly (yet remains < 100%) for tasks with outputs of the same modality as the anchor task, as seen for VQA in Fig.5(a). The % consistency increases from k=1 to k=2 by 12%, 8%, and 4% for UnifiedIO, OFA, and Kosmos-2 respectively, and then by smaller margins with increasing k. For the localization task, the % consistency increases from k=1 to k=1 by 12%, 1%, and 5% for UnifiedIO, OFA, and Kosmos-2 respectively, and then by smaller (or negative) margins with increasing k. However, we see almost no improvement in consistency for the text-to-image generation for easier contrast sets, implying that the unification of modalities within a model is a non-trivial challenge.
>
> We have rewritten the conclusions under the heading “Models are more accurate than consistent.” in Sec. 7.1 as follows:
>
> Unified-IO models feature well below the $x=y$ line while other models, i.e., OFA and Kosmos-2, lie close to the $x=y$ line. Unified-IO models are more accurate at the VQA and the localization tasks than the other models, without demonstrating a similar gain in consistency. This suggests that when the Unified-IO models make mistakes for one task they rarely make the same kind of mistakes on the other tasks, which is what would allow a model to achieve high consistency independently of accuracy. On the other hand, OFA and Kosmos-2 models show a tight correlation between accuracy and consistency.
>
> * **Scatter plots similar to 5(b) for k>1**:
> Thank you for the suggestion, we agree that this would be an important analysis to look at. We have included scatter plots similar to 5(b) for k=2, k=3, k=4, k=5 in Figure 13 in the Appendix (see revised pdf), and added the following lines to the discussion in Sec. 7.1 under the heading of “Models are more accurate than consistent.”:
>
> Additionally, we observe that the models gradually transition over the $x=y$ line with increasing $k$ i.e., easier contrast sets (see Fig. in Appendix). Notably, OFA$_{CON}$ consistently ranks the highest in terms of consistency at all $k$.
>
> * **Using likelihoods for text-to-image evaluation**:
> Thank you for the suggestion, we decided to analyze the calibration of the Unified-IO and OFA models for the text-to-image generation task using principles from [1] and report our results in Fig. 7 and Sec. 7.2 (see blue text) in the revised pdf. In summary,
>
> - We plot reliability maps for the image likelihoods generated by Unified-IO models, in comparison to the average CLIPScore value of the images, in Fig. 6 in the main text.
> - The reliability maps show that image likelihood scores are positively correlated with average CLIPScore over the entire range of likelihood scores. Any variations in the trend towards the tail end of the distribution can be attributed to small sample sizes (we also plot the cumulative density of the distribution in the lower half of Fig. 6).
> - Further, when we use a well-known calibration technique i.e. temperature calibration [1] on these scores, we see no significant improvements in the strength of the correlation or the linear regression fit (measured in terms of mean squared error) - see Fig. 11 in Appendix.
> - We also compare the distribution of likelihoods for various tasks and model sizes in Fig. 10 in the Appendix, and find that the distributions for pivot instances and contrast sets suffer high overlap for the text-to-image generation task for all model sizes, whereas we see increasing separation between the two distributions with model size for all other tasks.
> - All of this evidence suggests that the low values of cross-task consistency for the text-to-image generation task primarily stems from the complexity of the task itself as well as the gross misalignment between text and image output modalities in multimodal models.
>
> (..continued)

---

> > ### Author Response · Authors · 2023-10-25
> > **Response to requested questions and changes (continued...)**
> >
> > * **Standard error of the models**:
> > Our evaluation of the CocoCON benchmark is completely deterministic as we merely measure a model’s likelihood on a predefined output. We compare the likelihoods on various outputs and assign a value of 1 if the margins for two tasks have the same sign and 0 otherwise (see Equation 1 in Sec. 3). To report the consistency numbers in Table 3, we take a sum over these values. This process does not involve any sampling or random seeds and hence, we do not report any error bars from this evaluation. If you have any other suggestions in this regard, please let us know, and we will do our best to comply.
> >
> > * **Consistency-based Regularization**:
> > As we discuss in Sec. 3, optimizing for consistency can result in degenerate solutions where the accuracy is poor. Our proposal for rank-based regularization is an attempt towards addressing this open problem and providing an initial solution for achieving consistency without degrading accuracy. We do see large improvements i.e., 3% and 5% for the VQA and localization tasks respectively (compare rows G and M in Table 3) with a carefully selected balance between the regularization objective and cross-entropy loss. The potential incompatibilities between the definitions of consistency and accuracy prevent our proposed loss to fully close the gap in consistency while preserving the accuracy of the models.
> >
> > * **Additional Examples**:
> > We have added a series of additional examples to the Appendix.
> >
> > * **Limitations**:
> > Thank you for the suggestion! We have added a discussion on the limitations after the Conclusion in the revised pdf as follows:
> >
> > _Out-of-domain evaluation_: The CocoCON benchmark is the first step (to the best of our knowledge) towards measuring consistency across tasks in multimodal models. It is derived from the COCO dataset which has been widely used for pretraining multimodal models, hence, the data samples in CocoCON are in-domain for these models. In spite of CocoCON being sampled from the same distribution as the training data of such models, we are seeing significant inconsistencies across tasks, especially those of different output modalities. We expect that these inconsistencies will aggravate when the samples are drawn from an out-of-domain distribution. Our work does not conduct out-of-domain evaluation, nevertheless, our automated pipeline and evaluation framework provides the groundwork for preparing benchmarks for accomplishing the same in future research.
> >
> > _Additional Tasks_: Our evaluation and analysis are carried out on three tasks which may not represent the full spectrum of potential tasks in the computer vision and multimodal space. However, as we show in Fig. 2, this evaluation framework can be extended to more tasks. The use of captioning as a pivot was decided due to our focus on multimodal tasks. One can switch the pivot for a more computer-vision oriented task such as image classification and conduct evaluation on tasks like depth estimation.
> >
> > _Availability of cross-task annotations_: We chose the COCO dataset as a starting point for our benchmark dataset because it is a popular dataset that has been widely used for training and evaluation. The availability of annotations for multiple tasks served as a convenient starting point as it allowed us to create annotations for multiple tasks with the same pivot instance. However, cross-task consistency is defined for a pair of tasks and is evaluated independently of the other tasks. Hence, cross-task does not require multiple task annotations for the same pivot instance; one can have different pivot instances with annotations for a pair of tasks only, in order to scale this evaluation to more tasks. Further, many contemporary models super-specialize in one or more tasks such as Segment Anything for image segmentation, Kosmos-2 for localization, BLIP-2 for captioning and VQA. These models can be used to rapidly create annotations for a pair of tasks, and coupled with manual filtering, can be used for scaling to other datasets as well as tasks.
> >
> > * **Dataset Release & License**:
> >
> > This dataset will be released through a dedicated GitHub repository and popular dataset APIs such as HuggingFace Datasets. Since the images and caption/VQA/localization annotations are borrowed from the COCO dataset, we will be releasing them under the same license as COCO i.e. the Creative Commons license (see here: https://cocodataset.org/#termsofuse). The COCO Consortium does not own the copyright of the images, hence, we will not be distributing the images and will only release the Flickr URLs for the images. Additionally, we will also release the code for the automated pipeline described in Section 4.

---

> ### Author Response · Authors · 2024-01-04
> **Follow-Up**
>
> Dear Reviewer,
>
> We are writing to follow up regarding our revised manuscript. Kindly let us know if the changes in the revised version have addressed the issues raised in your review. We are happy to provide additional revisions or clarifications if needed. Thank you for your time!
>
> Best,
>
> Authors

---

> > ### Comment · Reviewer_pVDY · 2024-01-09
> >
> > Thank you for your rebuttal (and updating the paper) and apologies for the silence on my end. I was waiting for the other reviews to come in before getting back to you. My concerns have been adequately addressed and I will recommend acceptance.

---

> > > ### Author Response · Authors · 2024-01-10
> > >
> > > Thank you for the recommendation, Reviewer pVDY! Your feedback has helped us improve the work.

---

### Review · Reviewer_9JFw · 2023-10-01

**Summary Of Contributions:**

This paper offers a comprehensive analysis on the cross-task inconsistency in unified vision-language models. The authors bring attention to the prevalent issue of inconsistency in the existing models (i.e., Unified-IO, OFA), and introduce CocoCon, a benchmark framework employing contrast sets, tailored to measure inconsistency across four multi-modal tasks. The paper showcases the inconsistent behaviors manifested by leading vision-language models and proposes a novel consistency-based training objective, which enhances consistency while preserving model accuracy.

**Audience:**

Yes

**Claims And Evidence:**

Yes

**Requested Changes:**

### On the correlation between C1 and task accuracy.
	According to the equations presented in the paper, C1 is a pseudo accuracy of the target task calculated by utilizing the prediction result of the pivot task as the true values. If there is a model that perfectly executes the pivot task, then C1 for each task will be identical with the accuracy of each target task, showing a perfect x=y correlation. This equilibrium between C1 and the target task accuracy starts to distort as the accuracy of the pivot task decreases.

	If the pivot task and target task make the same errors, then the model would lie above the x=y line.

	Conversely, if the target task errs a sample that the pivot task does not, then the model would lie below the x=y line.

	If both the pivot task and target task err independently, the model would lie around the x=y line.

	If the pivot task completely guesses randomly, then the C1 value of the target task would converge to 0.5 regardless of the target accuracy.

	In essence, it might be more appropriate to understand that as accuracy increases, consistency also increases. For instance, the claim that an easier contrast set shows better consistency might be because it is naturally easier to discern, leading to a higher accuracy.

To conclude, further analysis and validation are required based on the assumption that accuracy and consistency naturally correlate.


### Observation on text-to-Image generation task
	For the Text-to-Image generation, almost all models, except for the GILL model, show a C1 around 50%. This seems to indicate that for this task, random guess is taking place. While the GILL model does show noteworthy numbers, its performance in all other tasks, including the pivot task, is significantly behind the baseline. This makes it ambiguous if the actual consistency has whether improved or not.

	The method employed by the authors to evaluate this task might be questionable. Given the diversity in images, measuring the probability of generating the correct image when given a contrast label is uncertain.

To conclude, there exist room for debate regarding whether the evaluation method the authors suggest is valid, and thus a deeper exploration is necessary regarding the assessment method employed for the Text-to-Image generation task.

**Strengths And Weaknesses:**

### Strengths
	The paper is well-structured and easy to follow, providing clarity for readers.

	Given the surge in recent unified model releases, there is a pressing need for evaluating task-wise consistency, making this paper particularly relevant and timely.

	The mechanism for creating the dataset is commendably automated, suggesting the feasibility of generating new contrast sets using other suitable datasets in the future.

	By employing the proposed framework, the authors adeptly illuminate the existence of inherent inconsistency in current unified models. The analysis outlines the tendencies displayed by such inconsistencies.

	A simple yet effective consistency-based loss demonstrates potential improvements in model consistency.

### Weaknesses
	The paper relies heavily on the COCO dataset. For the creation of the proposed contrast sets, it is imperative that captions, localization, and VQA labels exist for the same image. However, datasets fulfilling these criteria are scarce.

	There are certain ambiguities in understanding the correlation between accuracy and consistency. More details can be found in the "Requested Changes" section below.

---

> ### Author Response · Authors · 2023-10-25
> **Response to Requested Changes**
>
> Dear Reviewer,
>
> Thank you for taking the time and effort to review our manuscript and provide detailed feedback. We appreciate that you have found our work to be interesting, useful, and well-written! We have addressed your questions and concerns in the revised manuscript (see blue text), outlined the main changes below, and pointed to relevant sections or updated figures wherever applicable. Please let us know if you have any additional concerns, we will be happy to address those!
>
> ----------------------------------------------------------------------------------------
>
> * **On the Correlation between C1 and Accuracy**:
>
> We are grateful to the reviewer for their detailed description of the relation between accuracy and consistency! We have discussed this to some extent in Sec. 3 and Sec. 7.1, however, the reviewer has phrased it in a much more elegant manner that is immediately understandable to any reader. If the reviewer doesn’t mind, we would like to adapt the writing in the review and add it to our paper (in Section 3) for the benefit of readers of this work. Additionally, we have used the points mentioned by the reviewer to conduct a simulation of the various scenarios of model accuracy vs. consistency and provide a demonstration + discussion to validate these points in the Appendix (see Section A).
>
> * **Observation on text-to-image generation task**:
>
>  Thank you for the suggestion, we decided to analyze the calibration of the Unified-IO and OFA models for the text-to-image generation task using principles from [1] and report our results in Fig. 6 in the main text, Sec. 7.2 (see blue text) in the revised pdf and Fig. 10, 11 in the Appendix. In summary,
>
> - We plot reliability maps for the image likelihoods generated by Unified-IO models, in comparison to the average CLIPScore value of the images, in Fig. 6 in the main text.
> - The reliability maps show that image likelihood scores are positively correlated with average CLIPScore over the entire range of likelihood scores. Any variations in the trend towards the tail end of the distribution can be attributed to small sample sizes (we also plot the cumulative density of the distribution in the lower half of Fig. 6).
> - Further, when we use a well-known calibration technique i.e. temperature calibration [1] on these scores, we see no significant improvements in the strength of the correlation or the linear regression fit (measured in terms of mean squared error) - see Fig. 11 in Appendix.
> - This suggests that the model likelihoods can be used as a valid representation of model confidence in our proposed evaluation for text-to-image generation on the CocoCON benchmark.
> - We also compare the distribution of likelihoods for various tasks and model sizes in Fig. 10 in the Appendix and find that the distributions for pivot instances and contrast sets suffer high overlap for the text-to-image generation task for all model sizes, whereas we see increasing separation between the two distributions with model size for all other tasks.
> - All of this evidence suggests that the low values of cross-task consistency for the text-to-image generation task primarily stems from the complexity of the task itself as well as the gross misalignment between text and image output modalities in multimodal models.
>
> * **Reliance on the COCO Dataset**:
>
> We chose the COCO dataset as a starting point for our benchmark dataset because it is a popular dataset that has been widely used for training and evaluation. The availability of annotations for multiple tasks served as a convenient starting point as it allowed us to create annotations for multiple tasks with the same pivot instance. However, cross-task consistency is defined for a *pair of tasks* and is evaluated independently of the other tasks. Hence, cross-task does not require multiple task annotations for the same pivot instance; one can have different pivot instances with annotations for a pair of tasks only, in order to scale this evaluation to more tasks. Further, many contemporary models super-specialize in one or more tasks such as Segment Anything [2] for image segmentation, Kosmos-2 [3] for localization, BLIP-2 [4] for captioning, and VQA. These models can be used to rapidly create annotations for a pair of tasks, and coupled with manual filtering, can be used for scaling to other datasets as well as tasks. We have added this discussion point under Limitations (see after Conclusion) in the revised text.
>
> [1] Guo, Chuan, et al. "On calibration of modern neural networks." International conference on machine learning. PMLR, 2017.
>
> [2] Kirillov, Alexander, et al. "Segment anything." arXiv preprint arXiv:2304.02643 (2023).
>
> [3] Peng, Zhiliang, et al. "Kosmos-2: Grounding Multimodal Large Language Models to the World." arXiv preprint arXiv:2306.14824 (2023).
>
> [4]  Li, Junnan, et al. "Blip-2: Bootstrapping language-image pre-training with frozen image encoders and large language models." arXiv preprint arXiv:2301.12597 (2023).

---

> ### Author Response · Authors · 2024-01-04
> **Follow-up**
>
> Dear Reviewer,
>
> We are writing to follow up regarding our revised manuscript. Kindly let us know if the changes in the revised version have addressed the issues raised in your review. We are happy to provide additional revisions or clarifications if needed. Thank you for your time!
>
> Best,
>
> Authors

---

### Review · Reviewer_Wuuv · 2023-12-20

**Summary Of Contributions:**

1. The paper proposes a novel benchmark (CocoCon) and evaluation metrics for measuring cross task consistency in vision and language (V&L) models.
2. It selects image captioning as an anchor task and measures pairwise cross consistency with VQA, Image Generation and object detection.
3. CocoCon is built upon the COCO test set created by modifying the test instances for multiple tasks in small but semantically meaningful way to create contrast sets that changes the ground truth label. For each image it selects questions that have semantic overlap with the captions and generates contrast sets of the ground truth answers i.e similar but wrong answers. These contrast sets are then substituted in the caption, filtered out and used for evaluating consistency on other tasks.
4. The paper outline metrics for measuring if a model is consistent by ranking the original and contrast instances across tasks.
5. The results shows that although V&L models are accurate they are still inconistent across tasks and models trained on diverse tasks are more inconsistent with larger multi-task models that are more accurate being more consistent.

**Audience:**

Yes

**Broader Impact Concerns:**

None.

**Claims And Evidence:**

Yes

**Requested Changes:**

Mentioned above

**Strengths And Weaknesses:**

## Strengths

### Important Problem

- The paper addresses a significant issue: measuring cross-task consistency in multi-task V&L models.
- Given the trend of training a single V&L model for various tasks, ensuring consistent behaviour across all tasks is crucial.

### Benchmark

- The CocoCon benchmark utilizes images from the COCO dataset's test set.
- The data collection involves both automatic and manual filtering steps, ensuring a high-quality benchmark.
- The benchmark evaluates models across tasks with diverse output modalities, such as text, bounding boxes, and images, making it adaptable to other tasks.

### Extensive Experiments

- The paper conducts experiments on four leading V&L models, highlighting their inconsistencies.
- The experiments explore factors like model parameters, task nature and count, training task count, and contrast set quality.
- A rank correlation-based auxiliary training objective is proposed to enhance model consistency across all tasks.

### Presentation

- The paper is articulate and comprehensible.
- The data collection pipeline is clearly explained.

---

## Weaknesses

### Image Perturbation

- The benchmark primarily uses contrast sets derived from perturbing ground truth textual answers.
- To assess the model's understanding of images without language bias, the contrast sets could also be generated by perturbing images, as indicated in [1].

### Candidate Answers in Step 2

- Step 2 of the data collection pipeline generates alternatives (candidate answers) solely based on the ground truth answer, limiting contrast set diversity.
- Creating candidate answers for objects or visual elements mentioned in captions might yield more diverse and challenging contrast sets. The rationale behind relying solely on the ground truth answer needs clarification.

### Comparing Highly Similar Outputs

- How are the confidence scores of highly similar answers like "hair", "her hair" (id 1441) or "caution" "use caution" (id 1446) in the dataset compared? In such cases both the answers might be correct but the confidence score will slightly differ which might result in models performing lower on the benchmark when evaluted for cross consistency.

### Consistency between VQA and Captioning

- As the contrast sets are generated based upon the answers should'nt the VQA task be the anchor task?
- Furthermore, measuring cross consistency between Captioning and VQA seems to make no sense as the input remains the same for VQA (the image and question are unchanged), only the output label (answer in this case) is changed. Hence, if the model assigns a high likelihood to a certain answer correct or incorrect the cross consistency is solely dependent on the captioning. It seems one can evaluate cross consistency of VQA and Captioning based upon some relationship between the accuracies of the respective tasks. More explanation is required to demonstrate how the cross consistency is relevant between VQA and Captioning. The benchmark can either include VQA or Captioning as anchor task and other tasks can remain as such.

### Relationship with Accuracy

- Figure 8 in the appendix illustrates the relationship between consistency and accuracy without specifying the task. Is this relationship computed across all tasks? The details behind its computation remains are not in the paper. A dedicated figure for each pair of tasks would enhance the clarity and interpretation of the relationship.

### Extending to Other Tasks

- The paper could benefit from discussing how the benchmark and evaluation metrics might apply to other V&L tasks, such as referring expressions and visual entailment.

[1] Evaluating models' local decision boundaries via contrast sets.

---

> ### Author Response · Authors · 2024-01-03
> **Response to Requested Changes (1/n)**
>
> Dear Reviewer,
>
> Thank you for taking the time and effort to review our manuscript and provide feedback. We appreciate that you have found the topic of our work to be significant and our paper to be articulate! We have addressed your questions in our detailed response below and will make suitable changes in the revised manuscript wherever applicable:
> ***
> * **Image Perturbation**: The contrast sets in our benchmark dataset cover various semantic concepts as we demonstrate in Fig. 4 in the main text. While it is possible to perform suitable image perturbations algorithmically for the ‘Object’, ‘Animal’, ‘Location’ etc. categories using recently introduced text-guided image editing models [1,2], it may not be straightforward to do the same for semantic concepts in the more nuanced categories, for example, ‘standing’ → ‘running’ in ‘Action’, ‘Colgate’ → ‘Walgreens’ in ‘OCR’ and ‘three’ → ‘four’ in ‘Misc.’. This is because (a) text-to-image generation models generally struggle with challenges like spatial, and numerical composition and generation of text within images [3], and (b) current text-guided image editing methods only perform well for images containing one or two prominent subjects/objects, whereas the COCO images in CocoCON tend to contain multiple subjects and objects. Moreover, current text-guided image editing methods are not scalable. Creating text-based perturbations is relatively easier, which serves as a more scalable approach for extending our cross-task consistency framework to multiple tasks. Nevertheless, we agree that image-based perturbations are of significant interest as well; we will discuss the possibilities of future work in this direction in the paper, thank you for the suggestion.
>
> * **Candidate Answers in Step 2**:  The contrast set candidates in Step 2 were generated using the VQA question rather than the image caption to ensure they are challenging hard negatives for the VQA task. Additionally, it is simpler to generate VQA answers than to perform masked token prediction within a caption using decoder-only multimodal models like GPV2. To compare the diversity of contrast sets generated using VQA vs. those generated using captions, we performed a small experiment using the recent LlaVA 1.5 multimodal model [4]. We generate contrast set candidates for the examples demonstrated in Fig. 4 in the main text by replacing semantic concepts within captions. using the following prompt:
> ```
> Based on the image, write 5 different answers for filling in the blank in the following image caption: [caption with blank placeholder _ ]. For each answer, write your confidence in how well the answer fits into the caption while staying true to the image.
> ```
>
> The results are presented in the table below. Concepts replaced in the caption are italicized.:
> | Original Caption | CocoCON contrast sets (using VQA) | Contrast sets generated using caption (LlaVA 1.5) |
> | ------ | -------- | ----- |
> | A child in a bed with a striped sweater and a colorful *blanket*. | stuffed animal, pillow, teddy bear | stuffed animal, blanket, pillow, teddy bear |
> | A brown and *white* cat lying on the bed. | yellow, black, grey, orange | black, orange, tan, cream |
> | *Apples* are hanging from a tree that has hardly any leaves on it. | pear, olive, orange, cantaloupe | oranges, pears, peaches, plums |
> | Office space with *cat* on the television and work. | cartoon, squirrel, baseball, butterfly, bowling | - |
> | A man holding out to catch a *baseball*. | tee ball, basket ball, softball, football  |  frisbee, soccerball, tennis ball, golf ball |
> | A giant *colgate* clock sits on the shore next to water. | walgreens, billboard, state farm, wrigleys, yves saint lauren | coca-cola, clocktower, clockface |
>
> Overall, we did not see any significant difference in the diversity of the contrast sets generate using VQA vs. Caption. However, with LLM-based models like LlaVA, the data generation pipeline outlined in our paper can be simplified and we can remove the reliance on existing VQA annotations. Hence, we will add a discussion section on this to the main text and Appendix. Thank you for the useful suggestion!
>
> [1] Custom-Edit: Text-Guided Image Editing with Customized Diffusion Models
>
> [2] SINE: SINgle Image Editing with Text-to-Image Diffusion Models
>
> [3] HRS-Bench: Holistic, Reliable and Scalable Benchmark for Text-to-Image Models
>
> [4] Improved Baselines with Visual Instruction Tuning

---

> ### Author Response · Authors · 2024-01-07
> **Response to Requested Changes (2/n)**
>
> * **Comparing Highly Similar Outputs**: The issue of considering highly similar outputs for likelihood estimation is a subset of a broader, open challenge of aggregating over semantically equivalent outputs during the evaluation of free-form NLG. We were mindful of this issue when creating the CocoCON benchmark. To maintain uniformity in the evaluation setting, all contrast sets in CocoCON are made of noun, verb, adjective, or preposition words only. Additional pronouns like ‘her’ in ‘her hair’ (id: 1441) are neither replaced in the original caption nor added in the contrast sets if they are not significant semantic concepts. For instance, we retain the full phrase ‘motorcycles use caution’ because the text appears in the image (id: 1446). Some recent works propose methods for query/output expansion via mining, paraphrasing [5], multiple generations [6] etc., and then ensembling over a pool of similar outputs to get an aggregate score estimate. Since this is an active area of research, we leave their integration with cross-task consistency evaluation to future work. We will add this discussion to Limitations.
>
> * **Consistency between VQA and Captioning**:
>     - **Captioning as anchor task**: Image captioning serves as the best anchor task because it contains the most number of semantic concepts from the image that overlap with the outputs of a variety of other multimodal tasks. Besides VQA, localization, and text-to-image generation, some other tasks that can be easily compared to image captioning for cross-task consistency are referring expressions, hand-object interaction [7], object categorization [8], etc. On the other hand, VQA outputs generally do not contain overlaps with many other multimodal tasks. Hence, we choose image captioning as the anchor task. In our dataset generation pipeline, VQA simply serves as the means to collect contrast set candidates. As mentioned elsewhere in your review and our response (see 1/n), the contrast set generation pipeline can be modified to work with LLMs and captions; we will clarify this in the paper.
>
>     - **Cross-task Consistency**: In the case of cross-task consistency between VQA and captioning, the input remains the same for both tasks while the output is perturbed. For VQA, the image + question remains the same and the answer is changed. For image captioning, the image remains the same while the caption is changed. So the cross-task consistency between VQA and image captioning depends on whether the model assigns a higher likelihood to contrast set in both tasks or not. As we discuss in Sec. A. in the Appendix, for *any two given tasks*, the cross-task consistency across those tasks can be programmatically computed using their accuracies *only if* the model makes the same errors for both tasks. In that case, the amount of cross-task consistency is simply the difference between the accuracies of the tasks. If the model makes independent errors for both tasks, the cross-task consistency stays closer to 50% and if the model makes disjoint errors across the tasks, the cross-task consistency is generally low. The trends of cross-task consistency between VQA and Captioning follow the same intuition.
>
> * **Relationship with accuracy**: The results shown in Fig. 8 demonstrate results from *a simulation of model likelihoods* for two arbitrary tasks, as discussed in Sec. A. in the Appendix. We generate an array of likelihoods for ground truth and contrast set outputs for both tasks under different model conditions and then compute cross-task consistency using these likelihoods. We will clarify this in the image caption.
>
> * **Extending to Other Tasks**: Thank you for the suggestion, we will add this discussion to the paper:
>
> Our proposed framework for cross-task consistency evaluation can be easily extended to other V&L tasks. For instance, the consistency between image captioning and referring expression comprehension can be computed in the same way as done for localization. In this case, the semantic concepts appearing in the referring expression should appear in the caption as well, and one or more of those concepts are replaced with contrast sets. The ground truth bounding box of the referring expression can be used to estimate likelihoods (as done for localization in Sec. 4.3.). For computing consistency between image captioning and visual entailment, one can leverage the SNLI-VE dataset [9]. Those hypotheses that contain one or more semantic concepts that are also present in the caption, can be perturbed with contrast sets, and the likelihood of the ground truth label with this perturbed hypothesis can used for computing cross-task consistency.
>
> [5] Semantic Uncertainty: Linguistic Invariances for Uncertainty Estimation in Natural Language Generation
>
> [6] How Can We Know What Language Models Know?
>
> [7] Hand-Object Interaction Image Generation
>
> [8] GRIT: General Robust Image Task Benchmark
>
> [9] Visual Entailment: A Novel Task for Fine-Grained Image Understanding

---

> > ### Comment · Reviewer_Wuuv · 2024-01-07
> >
> > Thanks for the detailed responses and changes to the paper. I am convinced regarding the image perturbation, contrast sets curation and comparing highly similar outputs.  However, I still have doubts about the anchor task and cross task consistency between VQA and captioning.
> >
> > 1. Although, image captioning can serve as an anchor task as it contains the most number of semantic concepts from the image that overlap with the outputs, most semantic elements are not used while curating contrast sets. The only semantic elements used are the one in the VQA answers.
> >
> > 2. Furthermore regarding cross task consistency between VQA and captioning, the maximum likelihood of the model will remain the same (it will always result in same caption or answer). Therefore, the model predictions are always going to be same and it does not make sense to evaluate cross consistency between tasks where the predictions of the model are always same irrespective of which output is selected for likelihood comparison.

---

> > > ### Author Response · Authors · 2024-01-09
> > > **Response to Comment**
> > >
> > > Thank you for going through our responses and engaging with us about our work!
> > >
> > > (1) Since our work is the first study on cross-task consistency in V&L models (to the best of our knowledge), we built the evaluation framework and the CocoCON dataset with the hope that they will be extended to other tasks in subsequent research. We prepared CocoCON with the captioning task as the anchor so that future work can be built on top of our dataset and experiments. Further, at the time of creation of the CocoCON dataset, we did not have multimodal LLMs that could have enabled perturbation of any semantic concept within a caption, while being grounded to an image, at scale. Hence, we decided to use VQA as a simpler method for contrast set generation. Since that is no longer a constraint, we think that captioning as the anchor task is the best way forward.
> > >
> > > (2) We thought about this point again - and perhaps understood it better this time. You are right in saying that *technically* one does not require contrast sets to estimate cross-task consistency between VQA and captioning; it can be computed as
> > >
> > > $\frac{N(\text{samples where only one of the tasks is accurate}) +  N(\text{samples where both tasks are inaccurate as well as the errors are dissimilar})}{N(\text{all samples})}$.
> > >
> > > However, there are two issues with this:
> > > - 'accuracy' for image captioning is ill-defined. Evaluation metrics for captioning generally assign a continuous score that can't be used to decide whether a caption is objectively correct or incorrect, and
> > > - it is hard to compare a VQA output with a captioning output for similarity. Moreover, if the predicted caption does not contain any information about the semantic concept queried in the VQA sample (even if it appears in the ground truth), there is no way of knowing what the model 'thinks' about that visual concept.
> > >
> > > We overcome these issues by using contrast sets. By performing a pair-wise comparison between the ground truth and contrast set, we loosely redefine captioning accuracy in terms of their respective likelihoods. In the case of samples where the model is inaccurate at both, the captioning and VQA tasks, multiple contrast sets for each sample (as we have in CocoCON) let us analyze whether the errors are similar across both tasks.
> > >
> > > Let us know if this answers your doubts.

---

> ### Comment · Reviewer_Wuuv · 2024-01-09
>
> Thanks for the response. I agree with both the points mentioned above as it highlights the strengths and possible improvements for the contrast sets generation method and evaluating consistency between VQA and Captioning. Almost all of my issues have been addressed through the rebuttal and changes made to the paper.  After reading other reviews and responses I recommend accept.

---

> > ### Author Response · Authors · 2024-01-10
> >
> > Thank you for your review and recommendation, the feedback has improved our paper!

---

### Decision · Action_Editor_jK9H · 2024-02-01

**Recommendation:** Accept as is

**Comment:**

I agree with the reviewers to accept the paper.

I think the paper provides a very interesting benchmark and evaluation to increase our understanding of multimodal models, by evaluating the consistency of models across different multimodal tasks.

**Audience:**

yes! I think the paper is interesting for the vision and language/multimodal community as well as researchers from the ML community looking for benchmarks in consistency and reliability and insights in deep models.

**Claims And Evidence:**

Yes, the paper proposes a new benchmark for cross-task consistency in vision and language tasks. It provides the benchmark including metrics, an analysis, and an approach for training an improved model.